# Liu-Nagel phase diagrams in infinite dimension

## Giulio Biroli[1,2] and Pierfrancesco Urbani[1]

**1** Institut de physique théorique, Université Paris Saclay, CNRS, CEA, F-91191 Gif-Sur-Yvette
**2** Laboratoire de Physique Statistique, Ecole Normale Supérieure, PSL Research University, 24 rue Lhomond, 75005 Paris, France.

## Abstract

We study Harmonic Soft Spheres as a model of thermal structural glasses in the limit of infinite dimension. We show that cooling, compressing and shearing a glass lead to a Gardner transition and, hence, to a marginally stable amorphous solid as found for Hard Spheres systems. A general outcome of our results is that a reduced stability of the glass favors the appearance of the Gardner transition. Therefore using strong perturbations, e.g. shear and compression, on standard glasses or using weak perturbations on weakly stable glasses, e.g. the ones prepared close to the jamming point, are the generic ways to induce a Gardner transition. The formalism that we discuss allows to study general perturbations, including strain deformations that are important to study soft glassy rheology at the mean field level.

# 1   Introduction

When a liquid is cooled fast enough in such a way that crystallization is avoided, it enters in a
supercooled phase where upon further decreasing the temperature it freezes in an amorphous
solid phase, a glass [1]. This phenomenon has been investigated since the works by Adam
and Gibbs [2] that were aimed to clarify the thermodynamical nature of the glass transition.
Starting from the pioneering works by Kirkpatrick, Thirumalai and Wolynes [3–6], physicists
have realized that there exists a deep connection between the Adam-Gibbs picture of the glass
formation and the statistical physics of disordered systems such as spin glasses. The works of
Franz, Parisi, Mézard and Monasson [7–15] showed how to adapt the replica method, a very
powerful tool to study the properties of system with quenched disorder, to disorder-free Hamil-
tonians. This stream of ideas led finally to the exact description of the amorphous phases of
hard spheres in the limit of infinite dimension [16,17]. Hard sphere systems are good theoret-
ical models of colloidal glasses and have been studied in recent years to understand the critical
properties of the jamming transition. Remarkably, the mean field theory of hard sphere glasses
is able to correctly describe the criticality of jammed packings in three dimensions giving a very
accurate prediction of the critical exponents that appear at the jamming point. Furthermore,
the infinite dimensional solution of the hard sphere model has suggested that colloidal glasses
at very high pressure could undergo a new phase transition, the Gardner transition, that was
firstly found in models of spin glasses [18–21] and whose consequence in the structural glass
case are the object of a very intense research activity [22–26]. Beyond the Gardner point, hard
sphere glasses are predicted to be marginally stable: their properties are deeply affected by
non-trivial soft modes that drive strong non-linear elastic responses [25, 27–36].
The aim of the present work is to extend the analysis done for hard spheres to thermal glasses.
We shall present the phase diagram for elastic spheres in infinite dimension and thoroughly
study the properties of the amorphous solid phase. Our main purpose is to understand how
the properties of a glass evolve when external control parameters such as temperature, den-
sity and shear strain are changed. This is very reminiscent of the famous Liu-Nagel phase

diagram [37] in which it was shown that amorphous solids can be created or destablized varying temperature, density and stress. Anticipating some of our results, we show in Fig. 1 such a phase diagram obtained from the exact solution in the limit of infinite dimension. The difference, and the complication, compared to the original Liu-Nagel's one is that the properties of an amorphous solid depend on the temperature and density $(\widehat{T}_g, \widehat{\varphi}_g)$ at which the glass was formed, i.e. the point at which the super-cooled liquid falls out of equilibrium, and on the value of the temperature, pressure (or density) and shear strain that are applied. In consequence, a full phase diagram should actually contain five axis. These control parameters play different roles: the former are used to create the glass whereas the latter are the ones varied to probe and perturb the glass state. In order to simplify the presentation we therefore decided, both in Fig.1 and for explaining our results, to study the properties of amorphous solids by varying the temperature and density $(\widehat{T}_g, \widehat{\varphi}_g)$ at which they are formed and only one of the other control parameters at the time: temperature, density and shear strain. Fig. 1 shows an example of the results presented in this work, which corresponds to the following protocol: we apply a shear strain $\gamma$ to a glass prepared at $\gamma = 0$ and formed at a temperature and density corresponding to a point on the $(\widehat{T}_g, \widehat{\varphi}_g)$ plane (this means that the cooling rate is such that the glass transition takes place at $(\widehat{T}_g, \widehat{\varphi}_g)$). In the limit of infinite dimension glasses are well-defined and have an infinite life-time only below the dynamical line plotted in Fig.1. For each point $(\widehat{T}_g, \widehat{\varphi}_g)$ there are two critical values of the strain, a first one at which a Gardner transition takes place and then a second one corresponding to the yielding transition. By merging these points one obtains the Gardner and yielding critical surfaces shown in Fig.1.[1]

More details on the effect of the shear strain and analogous discussions for temperature and density changes are presented in the following sections. Before that, we present in Secs. 2,3,4,5,6 the analytical methods used and developed to study thermal glasses and a more straightforward derivation of the replica free-energy compared to previous works on hard-spheres [22–25]. A reader that wants directly to focus on the physical results can directly jump to Sec. 7 where we show the phase diagram of thermal glasses and we discuss how the properties of glasses evolve when external control parameters such as temperature, density and shear strain are changed.

## 2 Replica Theory for structural glasses: setting up the general formalism

We want to study amorphous states of particles interacting through a central potential $\hat{V}(r)$. Let us consider the case in which the system is at high temperature or low density in the so-called supercooled phase. On compressing or cooling the system, at some point the relaxation time will become very large and the system will not be able to equilibrate anymore [1]. It will thus fall out of equilibrium and become a glass. The point at which this freezing transition takes place depends strongly on the cooling and compression rate. The slower the compression rate or cooling rate are, the latter the system will fall out of equilibrium. In this way we can obtain different glasses that can be characterized by the inverse temperature $\beta_g$ and packing fraction[2] $\varphi_g$ at which they have fallen out of equilibrium. We will thus denote a generic glass state as $\alpha(\varphi_g, \beta_g)$. Once the supercooled liquid enters in the metastable glass state $\alpha$ we can think that on short timescales (shorter than the $\alpha$-relaxation time), the system is frozen in a portion of phase space that characterizes such glass state. However the equilibration of the

---

[1] Note that in the $\gamma = 0$ plane there is no Gardner transition because in our infinite dimensional analysis we are focusing on glasses formed at values of $(\widehat{T}_g, \widehat{\varphi}_g)$ above the Kauzmann transition.

[2] In general, the control parameter is the density $\rho$ but since we will deal with interacting spheres we will always use the packing fraction.

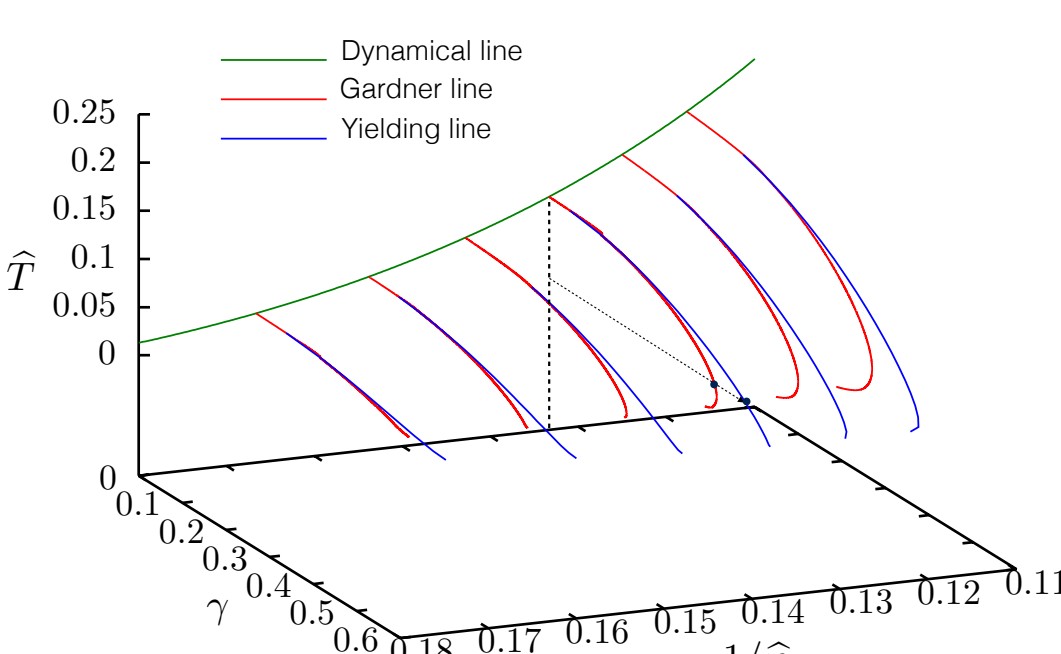

Figure 1: The mean field theory Liu-Nagel phase diagram. Glasses prepared in the $(1/\widehat{\varphi}, \widehat{T}, \gamma = 0)$ plane are strained. $\widehat{\varphi}$ and $\widehat{T}$ are scaled packing fraction and temperature These scalded variables are defined in the text in Eq. (18) and Eq. (42). Upon straining each glass undergoes first a Gardner transition and then a yielding instability. There are thus two surfaces: the most external one is the yielding surface and the internal is the Gardner one. Note that there is no Gardner transition in the plane at $\gamma = 0$. Indeed the plot we show here must be intended to be what is got when an equilibrium glass prepared in the glass region of the $\gamma = 0$ plane is strained. This can be also seen from Fig. (2) that gives the $\gamma = 0$ plane. Furthermore the yielding transition point must be intended to be only approximate because it is obtained through a 1RSB computation which is unstable beyond the Gardner surface.

system inside this restricted portion is still exponentially fast (it happens on the timescale of $\beta$-relaxation). Thus we can study the Boltzmann measure restricted to this metastable state. Let us consider a glass state prepared at $(\varphi_g, \beta_g)$ and then cooled or compressed up to the state point $(\varphi, \beta)$. We allow also the possibility to introduce a strain $\gamma$ with a deformation of the box in which the system is placed. We will denote the interaction potential in the strained box as $\hat{V}_\gamma(r)$. The free energy of such state is given by

$$f\left[\alpha(\varphi_g, \beta_g), \beta, \varphi, \gamma\right] = -\frac{1}{\beta N} \ln \int_{X \in \alpha(\varphi_g, \beta_g)} dX e^{-\beta \mathscr{V}_\gamma[X; \varphi]}, \tag{1}$$

where $\mathscr{V}_\gamma[X; \varphi] = \sum_{i<j} \hat{V}_\gamma(x_i - x_j)$ and $N$ is the system size. The notation $X \in \alpha(\varphi_g, \beta_g)$ denotes that we are summing up only configurations in phase space that belong to the ergodic component $\alpha(\varphi_g, \beta_g)$. The free energy (1) is called the Franz-Parisi potential and it has been introduced in spin glasses in [7] and it has been applied to structural glasses in [10, 38, 39]. For completeness, here we will discuss again this construction. For each given state point

$(\varphi_g, \beta_g)$ there are many possible glasses so that the free energy (1) is a random variable. In the thermodynamic limit we expect it to be self averaging and we can study its mean value

$$\overline{f\left[\alpha(\varphi_g,\beta_g),\beta,\varphi,\gamma\right]}^{\alpha(\varphi_g,\beta_g)} = -\frac{1}{\beta N}\overline{\ln\int_{X\in\alpha(\varphi_g,\beta_g)}\mathrm{d}X\,\mathrm{e}^{-\beta\mathcal{V}_\gamma[X;\varphi]}}^{\alpha(\varphi_g,\beta_g)}. \tag{2}$$

The average of the logarithm can be computed introducing replicas. If we define

$$W[\{\varphi_a,\beta_a,\gamma_a\}|\varphi_g,\beta_g] = -\frac{1}{N}\overline{\ln\prod_{a=1}^{s}\int_{X^{(a)}\in\alpha(\varphi_g,\beta_g)}\mathrm{d}X^{(a)}\mathrm{e}^{-\beta_a\mathcal{V}_{\gamma_a}[X^{(a)};\varphi_a]}}^{\alpha(\varphi_g,\beta_g)}, \tag{3}$$

where we have assumed that each replica has its own temperature, packing fraction and shear strain, the average free energy (2) is given by

$$\overline{\beta f\left[\alpha(\varphi_g,\beta_g),\beta,\varphi,\gamma\right]}^{\alpha(\varphi_g,\beta_g)} = \lim_{s\to 0}\frac{\partial}{\partial s}W[\{\varphi_a=\varphi,\beta_a=\beta,\gamma_a=\gamma\}|\varphi_g,\beta_g]. \tag{4}$$

The average over the different glassy states is given by

$$\overline{\prod_{a=1}^{s}\int_{X^{(a)}\in\alpha(\varphi_g,\beta_g)}\mathrm{d}X^{(a)}\mathrm{e}^{-\beta_a\mathcal{V}_{\gamma_a}[X^{(a)};\varphi_a]}}^{\alpha(\varphi_g,\beta_g)}$$
$$= \frac{1}{Z[\beta_g,\varphi_g]}\sum_{\alpha(\varphi_g,\beta_g)}\mathrm{e}^{-\beta_g N f\left[\alpha(\varphi_g,\beta_g)\right]}\prod_{a=1}^{s}\int_{X^{(a)}\in\alpha(\varphi_g,\beta_g)}\mathrm{d}X^{(a)}\mathrm{e}^{-\beta_a\mathcal{V}_{\gamma_a}[X^{(a)};\varphi_a]} \tag{5}$$

being

$$Z[\beta_g,\varphi_g] = \sum_{\alpha(\varphi_g,\beta_g)}\mathrm{e}^{-\beta_g N f\left[\alpha(\varphi_g,\beta_g)\right]} = \int \mathrm{d}f\,\mathrm{e}^{N\left(\Sigma(f)-\beta_g f\right)} \tag{6}$$

the partition function at temperature and packing fraction $(\varphi_g,\beta_g)$ and where $f[\alpha(\varphi_g,\beta_g)]$ is the free energy of the glass state $\alpha(\varphi_g,\beta_g)$. In this way the average over $\alpha(\varphi_g,\beta_g)$ is dominated by the equilibrium glassy states whose average free energy satisfies the relation

$$\frac{\mathrm{d}\Sigma(f)}{\mathrm{d}f} = \beta_g. \tag{7}$$

The function $\Sigma(f)$ is the configurational entropy or complexity [40]. It is well defined only at the mean field level and thus it is meaningful in the infinite dimensional limit that is the case we will study here. It is very useful to consider a biased partition function

$$Z_m[\beta_g,\varphi_g] = \sum_{\alpha(\varphi_g,\beta_g)}\mathrm{e}^{-\beta_g N m f\left[\alpha(\varphi_g,\beta_g)\right]} = \int \mathrm{d}f\,\mathrm{e}^{N\left(\Sigma(f)-\beta_g m f\right)}. \tag{8}$$

If we are able to compute $Z_m$ then we can compute

$$\Phi_m[\beta_g,\varphi_g] = -\frac{1}{N}\ln Z_m[\beta_g,\varphi_g], \tag{9}$$

from which we can reconstruct the configurational entropy. Indeed we can compute [8, 17]

$$\Sigma[m,\beta_g,\varphi_g] = m^2\frac{\partial}{\partial m}\left(\frac{1}{m}\Phi_m[\beta_g,\varphi_g]\right),$$
$$\beta_g f^*(m,\beta_g,\varphi_g) = \frac{\partial}{\partial m}\Phi_m[\beta_g,\varphi_g]. \tag{10}$$

From the parametric plot of $\Sigma$ as a function of $f^*$ we can reconstruct the configurational entropy $\Sigma(f)$ [8]. At this point we can introduce a generalized replicated free energy

$$W_m[\{\varphi_a, \beta_a, \gamma_a\}|\varphi_g, \beta_g, m] = \tag{11}$$

$$-\frac{1}{N}\ln\sum_{\alpha(\varphi_g,\beta_g)}e^{-m\beta_g Nf[\alpha(\varphi_g,\beta_g),\beta_g,\varphi_g,\gamma]}\prod_{a=1}^{s}\int_{X^{(a)}\in\alpha(\varphi_g,\beta_g)}dX^{(a)}e^{-\beta_a\mathscr{V}_{\gamma_a}[X^{(a)};\varphi_a]}. \tag{12}$$

From this expression it can be shown that

$$\overline{\beta f\left[\alpha(\varphi_g,\beta_g),\beta,\varphi,\gamma\right]}^{\alpha(\varphi_g,\beta_g)} = \lim_{s\to 0}\frac{\partial}{\partial s}W_1[\{\varphi_a=\varphi,\beta_a=\beta,\gamma_a=\gamma\}|\varphi_g,\beta_g,1]. \tag{13}$$

Moreover

$$\lim_{s\to 0}W_m[\{\varphi_a=\varphi,\beta_a=\beta,\gamma_a=\gamma\}|\varphi_g,\beta_g,m] = \Phi_m[\beta_g,\varphi_g]. \tag{14}$$

Finally, setting $m \neq 1$ gives us access to non-equilibrium states. Thus the basic object we want to compute is $W_m$. This can be computed by putting an infinitesimal coupling between $m$ replicas of the system that are at inverse temperature $\beta_g$ and packing fraction $\varphi_g$. In the glassy phase, at the mean field level, this coupling is enough to let all the replicas fall down inside the same glassy state [17] so that we can write

$$W_m[\{\varphi_a, \beta_a, \gamma_a\}|\varphi_g, \beta_g, m] = -\frac{1}{N}\ln\int\left(\prod_{a=1}^{n}dX^{(a)}\right)\exp\left[-\sum_{a=1}^{n}\beta_a\mathscr{V}_{\gamma_a}[X^{(a)};\varphi_a]\right], \tag{15}$$

where $n = m + s$, $\gamma_a = 0$, $\varphi_a = \varphi_g$ and $\beta_a = \beta_g$ for $a = 1,\dots,m$. In the next section we will compute $W_m$ in the high dimensional limit.

## 3  Derivation of the replicated free energy in presence of a shear strain in infinite dimension for a generic interaction potential

We want to derive the expression of $W_m[\{\gamma_a\}]$ in the limit of infinite dimension. This quantity has been already computed in [38] in the case of Hard Spheres and in [41] in the case of $s = 0$ and at zero shear strain for a generic interaction potential that is well behaved at large distances. Here we will present a simpler derivation that will allow us to generalize the calculation to the case $s > 0$ and most importantly when an external strain is applied to study elasticity and soft glassy rheology. We consider interaction potentials of the following form

$$-\beta\hat{V}(r) = -\widehat{\beta}\hat{v}\left(d\left(\frac{|r|}{\mathscr{D}}-1\right)\right), \tag{16}$$

where $d$ is the spatial dimension and $\mathscr{D}$ is the diameter of the spheres (or interaction range). Although we will construct the theory independently on the interaction potential, a simple example that we will use to obtain a quantitative phase diagram is the one of Harmonic Soft Spheres that interact through

$$\hat{V}_{\mathrm{HSS}}(r) = \frac{\epsilon}{2}\left(1-\frac{|r|}{\mathscr{D}}\right)^2\theta\left[1-\frac{|r|}{\mathscr{D}}\right]. \tag{17}$$

In this case we have

$$\hat{v}_{\mathrm{HSS}}(h) = \frac{\epsilon}{2}h^2\theta(-h), \qquad \widehat{\beta} = \frac{\beta}{d^2}. \tag{18}$$

We are also interested in the study of strained particle systems. In this case the box in which the system is placed is strained in one direction by an amount $\gamma$ and this deformation can be traced back into the effective interaction potential [38, 39, 42]

$$\hat{V}_\gamma(r) = \hat{V}(S(\gamma)r) \tag{19}$$

being $S(\gamma)$ a linear transformation defined by its effect on a $d$-dimensional vector $r$

$$[S(\gamma)r]_1 = r_1 + \gamma r_2 \,, \qquad [S(\gamma)r]_i = r_i \quad i = 2, \ldots, d \,. \tag{20}$$

In order to compute $W_m$ we need to consider $m+s$ systems or replicas of particles. The first $m$ replicas are composed by spheres with diameter $\mathscr{D}_g$ while in the last $s$ replicas the spheres have diameter $\mathscr{D} = \mathscr{D}_g(1 + \eta/d)$. The last $s$ replicas will thus be used to follow a glassy state planted at $(\varphi_g, \beta_g)$ to another state point $(\varphi, \beta, \gamma)$. In this way, the diameter $\mathscr{D}$ can be used to select the final packing fraction $\varphi$ and increasing $\eta$ will correspond to compress the system. Without losing generality we will set $\mathscr{D}_g = 1$ in what will follow. In the infinite dimensional limit the virial expansion [43] for $W_m$ can be truncated after the excess term [41, 43–45] and we have that

$$W_m[\rho] = -\frac{1}{N}\left[\int \mathrm{d}\underline{x}\rho(\underline{x})\big(1 - \ln\rho(\underline{x})\big) + \frac{1}{2}\int \mathrm{d}\underline{x}\mathrm{d}\underline{y}\rho(\underline{x})\rho(\underline{y})f(\underline{x}-\underline{y})\right], \tag{21}$$

where

$$\rho(\underline{x}) = \langle\sum_{i=1}^{N}\prod_{a=1}^{m+s}\delta\left(x_a - x_i^{(a)}\right)\rangle \tag{22}$$

and $x_i^{(a)}$ is a $d$-dimensional vector that gives the position of the sphere $i$ in replica $a$ [17]. Note that $\int \mathrm{d}\underline{x}\rho(\underline{x}) = N$. The first term that appears in the right hand side of (21) is called the *entropic* (or ideal gas) term while the second one is the *interaction* (or excess [43]) term. The function $f$ is the replicated Mayer function

$$f(\underline{x}-\underline{y}) = -1 + \prod_{a=1}^{n}\mathrm{e}^{-\beta_a\hat{V}_{\gamma_a}^{(a)}(x_a - y_a)} \tag{23}$$

and we have supposed that each replica has its own inverse temperature $\beta_a$ and its own shear strain deformation $\gamma_a$. Moreover we note that the interaction potential depends explicitly on the replica index since the first $m$ replicas have diameter $\mathscr{D}_g = 1$ while the last $s$ replicas have diameter $\mathscr{D}$.[3]

Although we will not use the Gaussian parametrization for $\rho(\underline{x})$ to obtain the results we will present, it is convenient to show it here since it could be used as an alternative route to derive the final equations [45]. The replicated system is translational and rotational invariant and we can introduce the displacement variables $u_a$ defined as

$$u_a = x_a - X \,, \qquad X = \frac{1}{n}\sum_a x_a \tag{24}$$

and $\rho(\underline{x})$ depends only on $\underline{u}$. We will denote $\rho(\underline{u})$ the distribution of the displacements. Thus

$$\int \mathscr{D}\underline{u}\rho(\underline{u}) = \rho \equiv N/V \,, \qquad \mathscr{D}\underline{u} = n^d\left(\prod_{a=1}^{n}\mathrm{d}^d u_a\right)\delta\left[\sum_{a=1}^{n}u_a\right]. \tag{25}$$

---

[3]We can also consider the most general case in which each replica has a different diameter $\mathscr{D}_a = \mathscr{D}_g(1 + \eta_a/d)$.

We can set up a Gaussian parametrization for $\rho(\underline{u})$ that is [18]

$$\rho(\underline{u}) = \frac{\rho n^{-d}}{(2\pi)^{(n-1)d/2} \left(\det A^{(n,n)}\right)^{d/2}} \exp\left[-\frac{1}{2} \sum_{a,b=1}^{n-1} \left[\left(A^{(n,n)}\right)^{-1}\right]_{ab} u_a \cdot u_b\right] \qquad (26)$$

and the matrix $A$ gives [46]

$$\langle u_a \cdot u_b \rangle = d A_{ab} . \qquad (27)$$

Due to translational invariance $A$ is a Laplacian matrix so that it satisfies

$$\sum_{a=1}^{n} A_{ab} = \sum_{b=1}^{n} A_{ab} = 0 . \qquad (28)$$

The matrix $A^{(n,n)}$ is the matrix that is obtained from $A$ by removing the last row and column.

In the infinite dimensional limit it has been shown in [18, 45, 46] that the correct scaling variable is

$$\alpha_{ab} = d^2 A_{ab} \qquad (29)$$

and the free energy can be rewritten in terms of a reduced packing fraction defined as $\widehat{\varphi}_g = 2^d \varphi_g / d$ being $\varphi_g = \rho \Omega_d / d$ where $\Omega_d$ is the surface of the unit sphere in $d$ dimensions and $\rho$ is the density of the system. It is also useful to define the matrix of the mean square displacements that is given implicitly in terms of $\alpha$

$$\Delta_{ab} = \alpha_{aa} + \alpha_{bb} - 2\alpha_{ab} . \qquad (30)$$

In order to compute the replicated free energy $W_m$ we need to evaluate the entropic and the interaction term that appear in Eq. (21). The entropic term in the high dimensional limit is given by [18, 45]

$$-\frac{1}{N} \int d\underline{x} \rho(\underline{x})\left(1 - \ln \rho(\underline{x})\right) = -\left(\text{const.} + \frac{d}{2} \log \alpha^{(n,n)}\right), \qquad (31)$$

where we have neglected irrelevant (for the sake of this work) constant terms. This expression can be equivalently derived assuming the Gaussian parametrization of Eq. (26).

The interaction term instead is more complicated. Its derivation has been done in complete generality for the Hard Sphere case. For generic interaction potentials instead, it has been obtained in [41] in the case of $s = 0$ and without any shear deformation. Here we will derive the expression for the interaction term in complete generality in an alternative and very compact way. In order to do this we first derive the interaction term as a function of $\Delta_{ab}$ in absence of any shear strain and then, using this result we extend our calculation to include its effect.

## 3.1 The interaction term in absence of a shear strain

We want to prove that in the infinite dimensional limit

$$
\begin{aligned}
I &= \frac{1}{2N} \int d\underline{x} d\underline{y} \rho(\underline{x}) \rho(\underline{y}) \left[-1 + \prod_{a=1}^{n} e^{-\beta_a \hat{V}^{(a)}(x_a - y_a)}\right] \\
&= -\frac{\widehat{\varphi}_g d}{2} \int_{-\infty}^{\infty} dh\, e^h \frac{d}{dh}\left[\exp\left[-\frac{1}{2} \sum_{a,b=1}^{n} \Delta_{ab} \frac{\partial^2}{\partial h_a \partial h_b}\right] \prod_{c=1}^{n} e^{-\widehat{\beta}_a \hat{v}(h_a)}\Big|_{\{h_c = h - \eta_c\}}\right] .
\end{aligned}
\qquad (32)
$$

We will not make any assumption on the exact form of $\rho(\underline{x})$ that by the way is fixed by the saddle point equation. Indeed, as it has been shown in [45], the replicated free energy, in the infinite dimensional limit, depends only on the first moments of the variables $\underline{x}$, as a consequence the central limit theorem. Using translational invariance we can first write

$$
d\underline{x} = d^d X \mathscr{D}\underline{u}, \qquad \mathscr{D}\underline{u} = n^d \left( \prod_{a=1}^n d^d u_a \right) \delta \left[ \sum_{a=1}^n u_a \right], \qquad X \equiv \frac{1}{n} \sum_{a=1}^n x_a, \qquad (33)
$$

and an analogous form holds for the measure $dy$. The field $\rho(\underline{x})$ depends only on the displacement variables $\underline{u}$ so that we will write $\rho(\underline{x}) \equiv \rho(\underline{u})$. In this way the interaction term can be rewritten as

$$
\begin{aligned}
I &= \frac{1}{2N} \int d\underline{x} \, d\underline{y} \, \rho(\underline{x})\rho(\underline{y}) \left[ -1 + \prod_{a=1}^n e^{-\beta_a \hat{V}^{(a)}(x_a - y_a)} \right] \\
&= \frac{1}{2\rho} \int d^d X \mathscr{D}\underline{u} \mathscr{D}\underline{v} \rho(\underline{u})\rho(\underline{v}) \left[ -1 + \prod_{a=1}^n \exp\left[ -\widehat{\beta}_a \hat{v}\left( -d\left( 1 - \frac{|X + u_a - v_a|}{1 + \eta_a/d} \right) \right) \right] \right] \\
&= \frac{\rho}{2} \int d^d X \mathscr{D}\underline{w} \tilde{\rho}(\underline{w}) \left[ -1 + \prod_{a=1}^n \exp\left[ -\widehat{\beta}_a \hat{v}\left( -d\left( 1 - \frac{|X + w_a|}{1 + \eta_a/d} \right) \right) \right] \right] \\
&\simeq \frac{\rho}{2} \int d^d X \mathscr{D}\underline{w} \tilde{\rho}(\underline{w}) \left[ -1 + \prod_{a=1}^n \exp\left[ -\widehat{\beta}_a \hat{v}\left( -d\left( 1 - |X + w_a|\left( 1 - \frac{\eta_a}{d} \right) \right) \right) \right] \right],
\end{aligned} \qquad (34)
$$

where

$$
\tilde{\rho}(\underline{w}) = \frac{1}{\rho^2} \int \mathscr{D}\underline{u} \rho(\underline{u})\rho(\underline{u} - \underline{w}), \qquad \int \mathscr{D}\underline{w} \tilde{\rho}(\underline{w}) = 1. \qquad (35)
$$

Translational invariance implies that

$$
\langle w_a \rangle \equiv \int \mathscr{D}\underline{w} \tilde{\rho}(\underline{w}) w_a = 0, \quad \langle w_a \cdot w_b \rangle \equiv \int \mathscr{D}\underline{w} \tilde{\rho}(\underline{w}) w_a \cdot w_b = \frac{2\alpha_{ab}}{d},
$$

$$
\sum_{a=1}^n \alpha_{ab} = \sum_{b=1}^n \alpha_{ab} = 0. \qquad (36)
$$

Let us now consider

$$
|X + w_a|^2 = |X|^2 + |w_a|^2 + 2X \cdot w_a \qquad (37)
$$

and look at the statistics of $|w_a|^2$. In the limit of infinite dimension the three terms on the RHS are all sums of a very large number of terms. This is because they are given in terms of a scalar product in $d$ dimension that can be expressed as a sum of $d$ terms. In consequence, one can take advantage of several simplifications induced by the central limit theorem. For example we have

$$
d \langle |w_a|^2 \rangle = 2\alpha_{aa}. \qquad (38)
$$

Moreover we have

$$
\langle X \cdot w_a \rangle = 0 \qquad (39)
$$

and

$$
d^2 \langle (X \cdot w_a)(X \cdot w_b) \rangle = 2|X^2|\alpha_{ab}, \qquad (40)
$$

where we have also used the rotational invariance. Note that only the first $n - 1$ of the $w_a$ vectors are independent due to translational invariance. This means that $dX \cdot w_a \sim \mathcal{O}(1)$, in

the infinite dimensional limit, becomes a Gaussian random variable $z_a$ such that $\langle z_a z_b \rangle = 2\alpha_{ab}$ and we can write

$$
\begin{aligned}
I &= \frac{\rho}{2} \int d^d X \int \left( \prod_{a=1}^{n} dz_a \right) \delta \left[ \sum_{a=1}^{n} z_a \right] \frac{\exp\left[ -\frac{1}{2} \sum_{a,b=1}^{n-1} \left[ \left(2\alpha^{(n,n)}\right)^{-1} \right]_{ab} z_a z_b \right]}{\left( (2\pi)^{n-1} \det(2\alpha^{(n,n)}) \right)^{1/2}} \\
&\quad \times \left[ -1 + \prod_{a=1}^{n} \exp\left\{ -\widehat{\beta}_a \hat{v}\left( -d\left( 1 - |X| \left( 1 + \frac{z_a}{d} + \frac{\alpha_{aa}}{d} \right) \left( 1 - \frac{\eta_a}{d} \right) \right) \right) \right\} \right] \\
&= \frac{\widehat{\varphi}_g d}{2} \int_{-\infty}^{\infty} dh\, e^h \int \left( \prod_{a=1}^{n} dz_a \right) \delta \left[ \sum_{a=1}^{n} z_a \right] \frac{\exp\left[ -\frac{1}{2} \sum_{a,b=1}^{n-1} \left[ \left(2\alpha^{(n,n)}\right)^{-1} \right]_{ab} z_a z_b \right]}{\left( (2\pi)^{n-1} \det(2\alpha^{(n,n)}) \right)^{1/2}} \\
&\quad \times \left[ -1 + \prod_{a=1}^{n} \exp\left[ -\widehat{\beta}_a \hat{v}\left( h - \eta_a + z_a + \alpha_{aa} \right) \right] \right] \\
&= \frac{\widehat{\varphi}_g d}{2} \int_{-\infty}^{\infty} dh\, e^h \int \left( \prod_{a=1}^{n} dz_a \right) \delta \left[ \sum_{a=1}^{n} z_a \right] \frac{\exp\left[ -\frac{1}{2} \sum_{a,b=1}^{n-1} \left[ \left(2\alpha^{(n,n)}\right)^{-1} \right]_{ab} z_a z_b \right]}{\left( (2\pi)^{n-1} \det(2\alpha^{(n,n)}) \right)^{1/2}} \\
&\quad \times \exp\left[ \sum_{a=1}^{n} \alpha_{aa} \frac{\partial}{\partial h_a} \right] \left[ -1 + \prod_{a=1}^{n} \exp\left[ -\widehat{\beta}_a \hat{v}\left( h_a - \eta_a + z_a \right) \right] \right] \Bigg|_{h_a = h},
\end{aligned}
\tag{41}
$$

where we have changed integration variable $|X| = 1 + h/d$ that produces the Jacobian factor $e^h$. Moreover $\rho\Omega_d \to \widehat{\varphi}_g d$ and [45, 46]

$$
\widehat{\varphi}_g = \varphi_g 2^d / d. \tag{42}
$$

At this point we can rewrite the Gaussian integral over the variables $z_a$ as a differential operator [46] to obtain

$$
\begin{aligned}
I &= \frac{\widehat{\varphi}_g d}{2} \int_{-\infty}^{\infty} dh\, e^h \exp\left[ \frac{1}{2} \sum_{a,b=1}^{n} (2\alpha_{ab}) \frac{\partial^2}{\partial h_a \partial h_b} + \sum_{a=1}^{n} \alpha_{aa} \frac{\partial}{\partial h_a} \right] \left[ -1 + \prod_{c=1}^{n} e^{-\widehat{\beta}_c \hat{v}(h_c - \eta_c)} \right] \Bigg|_{\{h_a = h\}} \\
&= \frac{\widehat{\varphi}_g d}{2} \int_{-\infty}^{\infty} dh\, e^h \exp\left[ -\frac{1}{2} \sum_{a,b=1}^{n} (\alpha_{aa} + \alpha_{bb} - 2\alpha_{ab}) \frac{\partial^2}{\partial h_a \partial h_b} \right] \left[ -1 + \prod_{c=1}^{n} e^{-\widehat{\beta}_c \hat{v}(h_c)} \right] \Bigg|_{\{h_a = h - \eta_a\}} \\
&= \frac{\widehat{\varphi}_g d}{2} \int_{-\infty}^{\infty} dh\, e^h \exp\left[ -\frac{1}{2} \sum_{a,b=1}^{n} \Delta_{ab} \frac{\partial^2}{\partial h_a \partial h_b} \right] \left[ -1 + \prod_{c=1}^{n} e^{-\widehat{\beta}_c \hat{v}(h_c)} \right] \Bigg|_{\{h_a = h - \eta_a\}} \\
&= -\frac{\widehat{\varphi}_g d}{2} \int_{-\infty}^{\infty} dh\, e^h \frac{d}{dh} \exp\left[ -\frac{1}{2} \sum_{a,b=1}^{n} \Delta_{ab} \frac{\partial^2}{\partial h_a \partial h_b} \right] \left[ \prod_{c=1}^{n} e^{-\widehat{\beta}_c \hat{v}(h_c)} \right] \Bigg|_{\{h_a = h - \eta_a\}},
\end{aligned}
\tag{43}
$$

where we have heavily used integration by parts. If we consider $\hat{v} = \hat{v}_{\mathrm{HSS}}$ and we take the hard sphere limit

$$
e^{-\widehat{\beta}\hat{v}_{\mathrm{HSS}}(h)} \to \theta(h), \tag{44}
$$

we get back the same result of [38, 46].

## 3.2 The interaction term in presence of a shear strain

Here we want to generalize the previous calculation to the case in which we add a shear strain to the system. We want to compute

$$
\begin{aligned}
I &= \frac{1}{2N}\int \mathrm{d}\underline{x}\mathrm{d}\underline{y}\,\rho(\underline{x})\rho(\underline{y})\left[-1+\prod_{a=1}^{n}\mathrm{e}^{-\beta_a\hat{V}(S(\gamma_a)(x_a-y_a))}\right]\\
&= \frac{1}{2\rho}\int \mathrm{d}^d X\mathscr{D}\underline{u}\mathscr{D}\underline{v}\rho(\underline{u})\rho(\underline{v})\left[-1+\prod_{a=1}^{n}\exp\left[-\widehat{\beta}_a\hat{v}\left(-d\left(1-\frac{|S(\gamma_a)(X+u_a-v_a)|}{1+\eta_a/d}\right)\right)\right]\right]\\
&= \frac{\rho}{2}\int \mathrm{d}^d X\mathscr{D}\underline{w}\,\tilde{\rho}(\underline{w})\left[-1+\prod_{a=1}^{n}\exp\left[-\widehat{\beta}_a\hat{v}\left(-d\left(1-\frac{|S(\gamma_a)(X+w_a)|}{1+\eta_a/d}\right)\right)\right]\right]\\
&\simeq \frac{\rho}{2}\int \mathrm{d}^d X\mathscr{D}\underline{w}\,\tilde{\rho}(\underline{w})\left[-1+\prod_{a=1}^{n}\exp\left[-\widehat{\beta}_a\hat{v}\left(-d\left(1-|S(\gamma_a)(X+w_a)|\left(1-\frac{\eta_a}{d}\right)\right)\right)\right]\right].
\end{aligned}
\tag{45}
$$

We then consider

$$
\begin{aligned}
|S(\gamma_a)(X+w_a)|^2 = |X+w_a|^2 &+ 2\gamma_a\left(x_1 x_2 + x_1 w_a^{(2)} + x_2 w_a^{(1)} + w_a^{(1)}w_a^{(2)}\right)\\
&+ \gamma_a^2\left(x_2^2 + 2x_2 w_a^{(2)} + \left(w_a^{(2)}\right)^2\right),
\end{aligned}
\tag{46}
$$

where $w_a^{(1)}$ and $w_a^{(2)}$ are the first and second component of the vector $w_a$. Using the same line of reasoning of Eq. (39) and Eq. (40) and taking only the leading contributions, we can show that in the large $d$ limit we can write

$$
|S(\gamma_a)(X+w_a)| \simeq |X| + \hat{x}\cdot w_a + \frac{1}{2}\frac{|w_a|^2}{|X|} + \gamma_a\frac{x_1 x_2}{|X|} + \frac{1}{2}\gamma_a^2\frac{x_2^2}{|X|},
\tag{47}
$$

where $\hat{x} = X/|X|$. We can now go to polar coordinates to write

$$
x_1 = |X|f_1(\theta_d),\qquad x_2 = |X|f_2(\theta_d),
\tag{48}
$$

being $\theta_d$ the polar angle in $d$ dimension. We have

$$
|S(\gamma_a)(X+w_a)| \simeq |X| + \hat{x}\cdot w_a + \frac{1}{2}\frac{|w_a|^2}{|X|} + \gamma_a|X|f_1(\theta_d)f_2(\theta_d) + \frac{1}{2}\gamma_a^2|X|f_2(\theta_d)^2.
\tag{49}
$$

In the appendix A we show that

$$
\lim_{d\to\infty}\frac{d^{\frac{a+b}{2}}}{\Omega_d}\int \mathrm{d}\theta_d f_1^a(\theta_d)f_2^b(\theta_d) = \begin{cases} 0 & \text{if } a \text{ or } b \text{ are odd}\\ (a-1)!!(b-1)!! & \text{otherwise,}\end{cases}
\tag{50}
$$

so that $\sqrt{d}f_i(\theta_d)$ are Gaussian random variables with zero mean and unit variance. Changing again integration variable $|X| = 1 + h/d$ we can write

$$
-d\left(1-|S(\gamma_a)(X+w_a)|\left(1-\frac{\eta_a}{d}\right)\right) = h - \eta_a + z_a + \alpha_{aa} + \gamma_a\sigma_1\sigma_2 + \frac{1}{2}\gamma_a^2\sigma_2^2,
\tag{51}
$$

where $\sigma_1$ and $\sigma_2$ are Gaussian random variables with zero mean and unit variance. We thus have

$$
\begin{aligned}
I &\simeq \frac{\rho}{2} \int d^d X \mathcal{D}\underline{w}\, \tilde{\rho}(\underline{w}) \left[ -1 + \prod_{a=1}^{n} \exp\left[ -\widehat{\beta}_a \hat{v}\left( -d\left(1 - |S(\gamma_a)(X+w_a)|\left(1-\frac{\eta_a}{d}\right)\right)\right)\right]\right] \\
&\simeq \frac{\widehat{\varphi}_g d}{2} \int_{-\infty}^{\infty} dh\, e^h \int \frac{d\sigma_1 d\sigma_2}{2\pi} e^{-(\sigma_1^2+\sigma_2^2)/2} \int \left(\prod_{a=1}^{n} dz_a\right) \\
&\quad \times \delta\left[\sum_{a=1}^{n} z_a\right] \frac{\exp\left[-\frac{1}{2}\sum_{a,b=1}^{n-1}\left[(2\alpha^{(n,n)})^{-1}\right]_{ab} z_a z_b\right]}{\left((2\pi)^{n-1}\det(2\alpha^{(n,n)})\right)^{1/2}} \\
&\quad \times \left[ -1 + \prod_{a=1}^{n} \exp\left[-\widehat{\beta}_a \hat{v}\left(h - \eta_a + z_a + \alpha_{aa} + \gamma_a \sigma_1 \sigma_2 + \frac{1}{2}\gamma_a^2 \sigma_2^2\right)\right]\right].
\end{aligned}
\tag{52}
$$

Using the differential representation for Gaussian integrals we get

$$
\begin{aligned}
I &\simeq \frac{\widehat{\varphi}_g d}{2} \int_{-\infty}^{\infty} dh\, e^h \int \frac{d\sigma_1 d\sigma_2}{2\pi} e^{-(\sigma_1^2+\sigma_2^2)/2} \\
&\quad \times \exp\left[\sum_{a=1}^{n}\left(\gamma_a \sigma_1\sigma_2 + \frac{1}{2}\gamma_a^2\sigma_2^2\right)\frac{\partial}{\partial h_a} - \frac{1}{2}\sum_{a,b=1}^{n}\Delta_{ab}\frac{\partial^2}{\partial h_a \partial h_b}\right]\left[-1+\prod_{a=1}^{n}e^{-\widehat{\beta}_a\hat{v}(h_a)}\right]\Bigg|_{h_a=h-\eta_a} \\
&= \frac{\widehat{\varphi}_g d}{2} \int_{-\infty}^{\infty} dh\, e^h \int \frac{d\zeta}{\sqrt{2\pi}} e^{-\zeta^2/2} \exp\left[-\frac{1}{2}\sum_{a,b=1}^{n}\left(\Delta_{ab} + \frac{\zeta^2}{2}(\gamma_a-\gamma_b)^2\right)\frac{\partial^2}{\partial h_a \partial h_b}\right] \\
&\quad \times \left[-1+\prod_{a=1}^{n}e^{-\widehat{\beta}_a\hat{v}(h_a)}\right]\Bigg|_{h_a=h-\eta_a} \\
&= -\frac{\widehat{\varphi}_g d}{2} \int_{-\infty}^{\infty} dh\, e^h \int \frac{d\zeta}{\sqrt{2\pi}} e^{-\zeta^2/2} \frac{d}{dh}\exp\left[-\frac{1}{2}\sum_{a,b=1}^{n}\left(\Delta_{ab} + \frac{\zeta^2}{2}(\gamma_a-\gamma_b)^2\right)\frac{\partial^2}{\partial h_a \partial h_b}\right] \\
&\quad \times \left[\prod_{a=1}^{n}e^{-\widehat{\beta}_a\hat{v}(h_a)}\right]\Bigg|_{h_a=h-\eta_a}.
\end{aligned}
\tag{53}
$$

If we take the hard sphere limit of Eq. (44) we get back the replicated free energy that has been studied in [38,39,47]. However the main advantage here is that this formula can be applied to soft potential and thus it can be used to study soft glassy rheology.

### 3.3 The final expression of the replicated free energy

We can finally summarize our result for the expression of the replicated free energy in the case of a strained system. We got

$$
\begin{aligned}
W_m[\{\varphi_a, \beta_a, \gamma_a\}|\varphi_g, \beta_g, m] &= -\Bigg[ \text{const} + \frac{d}{2}\log\alpha^{(n,n)} - \frac{\widehat{\varphi}_g d}{2}\int_{-\infty}^{\infty} dh\, e^h \int \frac{d\zeta}{\sqrt{2\pi}} e^{-\zeta^2/2} \\
&\quad \times \frac{d}{dh}\exp\left[-\frac{1}{2}\sum_{a,b=1}^{n}\left(\Delta_{ab}+\frac{\zeta^2}{2}(\gamma_a-\gamma_b)^2\right)\frac{\partial^2}{\partial h_a\partial h_b}\right]\left[\prod_{a=1}^{n}e^{-\widehat{\beta}_a\hat{v}(h_a)}\right]\Bigg|_{h_a=h-\eta_a}\Bigg],
\end{aligned}
\tag{54}
$$

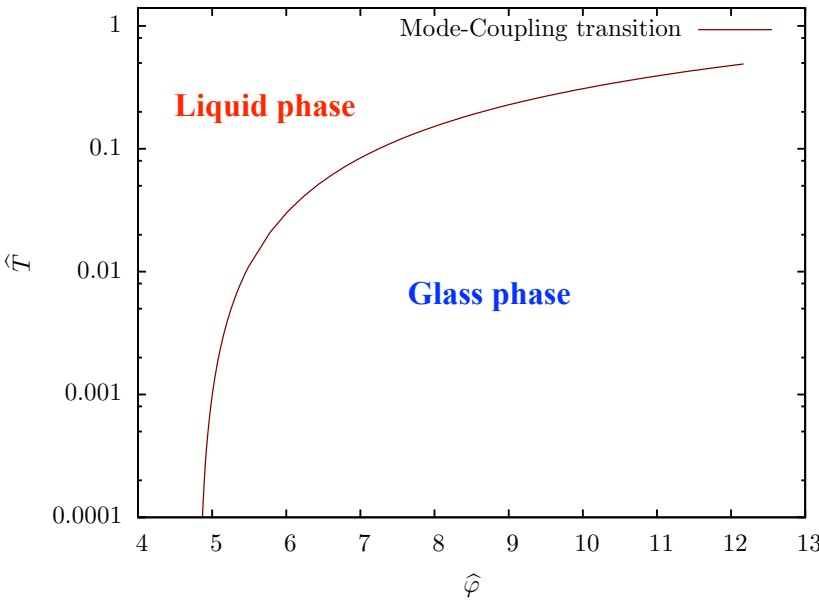

Figure 2: Phase Diagram of Harmonic Soft Spheres in the infinite dimensional limit. At zero temperature, the line converges to the Mode Coupling transition point of Hard Spheres at $\widehat{\varphi}_{MCT} \sim 4.8$.

where $n = m + s$, $\gamma_a = 0$, $\varphi_a = \varphi_g$ and $\beta_a = \beta_g$ for $a = 1, \dots, m$ and

$$\Delta_{ab} = \alpha_{aa} + \alpha_{bb} - 2\alpha_{ab} . \tag{55}$$

At this point we are equipped to study both where glassy states appear in the phase diagram and how they behave when the are compressed, cooled or strained.

## 4 The planted system: the dynamical transition

We would like first to study where metastable glassy states appear in the $(\widehat{T}_g, \widehat{\varphi}_g)$ plane. Since here we are not following the evolution of the system neither in compression, temperature or shear strain, we can directly set $s = 0$ in (15) and study $\Phi_m\left[\widehat{\beta}_g, \widehat{\varphi}_g\right]$ that is given by

$$\Phi_m\left[\widehat{\beta}_g, \widehat{\varphi}_g\right] = -\left[\text{const.} + \frac{d}{2}\log\alpha^{(m,m)} \right.$$
$$\left. -\frac{\widehat{\varphi}_g d}{2}\int_{-\infty}^{\infty} dh\, e^h \frac{d}{dh}\exp\left[-\frac{1}{2}\sum_{a,b=1}^{m}\Delta_{ab}\frac{\partial^2}{\partial h_a \partial h_b}\right]\left[\prod_{a=1}^{m} e^{-\widehat{\beta}_g \hat{v}(h_a)}\right]\Bigg|_{h_a=h}\right]. \tag{56}$$

The point in which glassy states appear in the Boltzmann distribution is signaled by a non trivial saddle point solution for $\Delta_{ab}$ in the limit $m \to 1$ that gives back the equilibrium measure [8,17]. In the simplest situation, having the $m$ replicas correlated means that the saddle point solution for $\Delta_{ab}$ assumes the form

$$\Delta_{ab} = \Delta_g(1 - \delta_{ab}) \tag{57}$$

that is the so called 1RSB ansatz. The validity of this solution can be checked a posteriori by looking at its local stability [18]. Plugging the 1RSB ansatz into (56) and taking the variational

equation we get the following saddle point equation

$$\frac{1}{\widehat{\varphi}} = \frac{\Delta_g}{m-1} \frac{\partial}{\partial \Delta_g} e^{-\Delta_g/2} \int_{-\infty}^{\infty} dh\, e^h \left[ 1 - g_{\Delta_g}^m(1,h;\widehat{\beta}) \right], \tag{58}$$

where

$$g_{\Delta_g}(1,h;\widehat{\beta}) = \gamma_{\Delta_g} \star e^{-\widehat{\beta}\hat{v}(h)} = \int_{-\infty}^{\infty} \frac{dz}{\sqrt{2\pi\Delta_g}} \exp\left[ -\frac{z^2}{2\Delta_g} - \widehat{\beta}\hat{v}(h-z) \right], \tag{59}$$

being $\gamma_\sigma$ a normalized Gaussian with variance $\sigma$. Let us introduce $A_g = \Delta_g/2$. Then, the saddle point equation can be rewritten as

$$\frac{1}{\widehat{\varphi}} = \frac{A_g}{m-1} \frac{\partial}{\partial A_g} \int_{-\infty}^{\infty} dh\, e^h \left[ 1 - g_{2A_g}^m(1,h+A_g;\widehat{\beta}) \right] \equiv \mathscr{F}_m\left(\widehat{\beta};A_g\right). \tag{60}$$

At fixed $\widehat{\beta}$ this equation admits a non trivial solution only for

$$\widehat{\varphi} \geq \frac{1}{\max_{A_g} \mathscr{F}_m\left(\widehat{\beta},A_g\right)} \equiv \varphi_d(m;\widehat{\beta}). \tag{61}$$

To obtain the equilibrium dynamical transition line in the $(\widehat{T},\widehat{\varphi})$ plane we have to take the limit $m \to 1$ so that we have

$$\widehat{\varphi}_{MCT}(\widehat{T}) = \varphi_d(1,\widehat{\beta}). \tag{62}$$

Note that in the limit $m \to 1$ we have

$$\mathscr{F}_1\left(\widehat{\beta};A_g\right) = -A_g \int_{-\infty}^{\infty} dh\, e^h \left[ \frac{d}{dA_g} g_{2A_g}(1,h+A_g;\widehat{\beta}) \right] \ln g_{2A_g}(1,h+A_g;\widehat{\beta}). \tag{63}$$

In Fig. 2 we show the dynamical line of Harmonic soft spheres obtained using

$$\hat{v}(h) \to \hat{v}_{\text{HSS}}(h) = \frac{h^2}{2}\theta(-h). \tag{64}$$

Finally the equations above give back well known results when the hard sphere limit is taken [17, 18, 45].

# 5 Non equilibrium states and off equilibrium dynamics: the real replica approach

Beyond the dynamical or mode coupling transition point, the system can be in different glasses. Indeed, if it is at equilibrium, the Boltzmann measure is dominated by glassy states that satisfy Eq. (7). However beyond equilibrium states, there are non equilibrium ones that can be studied as well. This can be done playing with the parameter $m$ in such a way that it biases the Boltzmann measure in Eq. (8) so that the dominant metastable states will be different from the true equilibrium ones. Indeed, at fixed $m$ the partition function in Eq. (8) is dominated by the states whose internal free energy satisfies

$$\frac{d\Sigma(f)}{df} = m\beta_g. \tag{65}$$

In this way we can study directly Eq. (8) to obtain both the equilibrium and off-equilibrium properties of the glass phase. Eq. (8) can be computed usign a fullRSB ansatz as we show in Appendix C and the corresponding saddle point equations can be easily reduced to Eq. (58) once computed on a replica symmetric solution. However the advantage of computing the fullRSB solution is that it gives directly access to the stability of the 1RSB saddle point.

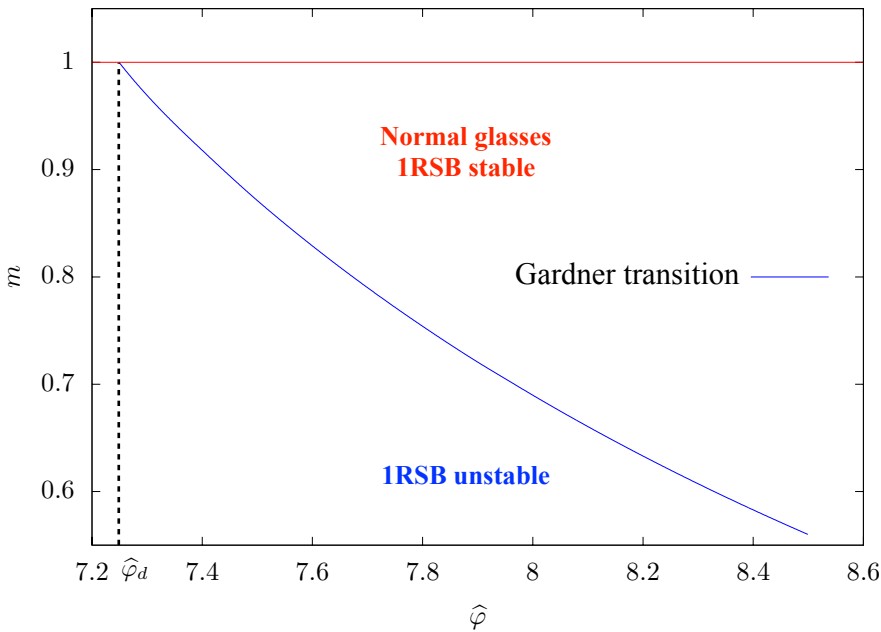

Figure 3: The Gardner transition line in the $(\widehat{\varphi}, m)$ plane for harmonic soft spheres at $\widehat{T} = 0.1$. Below this line the 1RSB solution is unstable. In the unstable phase there are two possibilities depending on wether the exponent parameter $\lambda(m)$ defined in Sec 7.A of [46] is greater or smaller than $m$. If $\lambda(m) > m$, below the Gardner line there could be a set of fullRSB marginally stable glassy states. Otherwise there are no glassy states at all [48].

## 5.1 The Gardner transition for the planted glass state

The stability of the 1RSB solution at the saddle point level can be checked by computing the replicon eigenvalue. Following exactly the same steps of Sec. 12 of [46] and using the fullRSB equations of Appendix C we can show that the condition for the instability of the 1RSB solution is given by

$$0 = -1 + \frac{\widehat{\varphi}_g}{2} e^{-\Delta_g/2} \int_{-\infty}^{\infty} dh \, e^h g_{\Delta_g}^m(1, h; \widehat{\beta}_g) \left[ \Delta_g \frac{d^2}{dh^2} \ln g_{\Delta_g}(1, h; \widehat{\beta}_g) \right]^2 . \qquad (66)$$

In Fig. 3 we plot the Gardner line in the $(m, \widehat{\varphi}_g)$ having fixed the temperature.

In the limit $\widehat{\beta}_g \to \infty$ this equation gives back the phase diagram of Hard Spheres [18]. Below the instability line there are three possibilities. The first one is that no glassy states exist. In order to test this possibility we must compute the exponent parameter $\lambda$ defined in Sec. 7.A of [46]. Here we do not extend this calculation to the soft sphere case since it can be easily done following the hard sphere case. The second possibility is that the Gardner transition is indeed a continuous transition towards a phase where a solution with a finite number of replica symmetry breaking steps is stable. This situation although not impossible is very baroque and we do not expect it in a generic case. The last possibility, which will be our working hypothesis throughout this work, is that the transition is towards a phase where replica symmetry is broken in a continuous way and glassy states are marginally stable. It has been shown that this is the case in the Hard Sphere limit [39, 46].

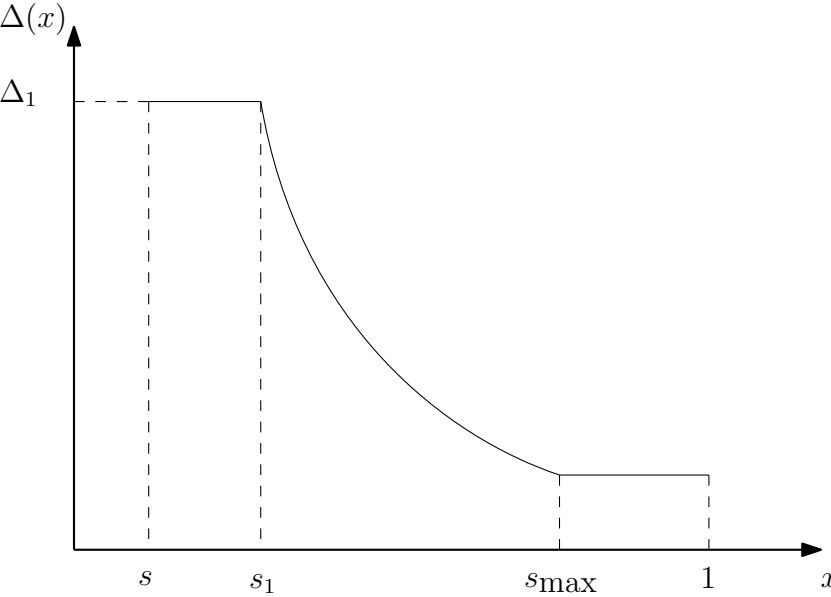

Figure 4: The generic profile of $\Delta(x)$ within the fullRSB solution.

# 6 The potential method: following adiabatically glassy states under external perturbations

In the previous sections we have studied in detail the replicated partition function (8) that gives access to glassy states. Here we want to see how they behave when an external perturbation is switched on. We are interested in three main situations: the system is compressed, cooled or strained. The formalism that gives access to the properties of glasses prepared at $(\widehat{\varphi}_g, \widehat{\beta}_g)$ and then followed up to $(\widehat{\varphi}, \widehat{\beta}, \gamma)$ is the Franz-Parisi potential that has been discussed in the Sec. 2. This can be obtained by computing Eq. (12) and Eq. (13).

## 6.1 The state-following calculation for a general interaction potential: fullRSB equations

The computation of Eq. (12) and Eq. (13) can be done by extending the formalism developed in [39]. Here we will not give all the technical steps and we will show only the final result. The computation of Eq. (12) and Eq. (13) can be done only by assuming a replica symmetry breaking scheme. Here we will assume that the $m$ replicas that represent the planted state are in a 1RSB stable glass phase and we will consider a fullRSB scheme for the replicas that describe the glass once followed in parameter space. This means that the form of the matrix $\Delta_{ab}$ is given by a symmetric matrix whose components are

$$
\begin{aligned}
&\Delta_{ab} = \Delta_g(1-\delta_{ab}) \qquad a,b = 1,\dots m; \\
&\Delta_{ab} = \Delta_r \qquad a = 1,\dots m; \ b = m+1,\dots,m+s \\
&\Delta_{ab} \to \{0, \Delta(x)\} \qquad a,b = m+1,\dots,m+s\,.
\end{aligned}
\tag{67}
$$

Here we will not describe in details the fullRSB parametrization of the sector of slave replicas since it can be found in [39, 46]. The general form of the function $\Delta(x)$ is plotted in Fig. (4) and we assume $\Delta(x) = 0$ for $x < s$. Using this parametrization and assuming that all the slave $s$ replicas have the same packing fraction and strain, namely that $\widehat{\beta}_a = \widehat{\beta}$, $\eta_a = \eta$, $\gamma_a = \gamma$ for

all $a = m+1, \ldots, m+s$, the replicated free energy is given by

$$-W_m[\{\widehat{\varphi}_a, \beta_a = \widehat{\beta}, \gamma_a\}|\widehat{\varphi}_g, \widehat{\beta}_g, m] = \text{const} + \frac{d(m-1)}{2} \log \Delta_g + \frac{d}{2} \log[m(\langle\Delta\rangle + s\Delta_1) + ms\Delta_f$$
$$+ s\Delta_g] + \frac{d}{2}(s-1)\log\langle\Delta\rangle - s\frac{d}{2}\int_s^1 \frac{\mathrm{d}y}{y^2}\log\left(\frac{\langle\Delta\rangle + [\Delta](y)}{\langle\Delta\rangle}\right)$$
$$- \frac{d\widehat{\varphi}_g}{2}\int_{-\infty}^{\infty} \frac{\mathrm{d}\zeta}{\sqrt{2\pi}}e^{-\zeta^2/2}\int_{-\infty}^{\infty}\left[\mathrm{d}h\,e^h\left\{1 - g_{\Delta_g}^m\left(1, h + \Delta_g/2; \widehat{\beta}_g\right)\right.\right.$$
$$\left.\left. \times \gamma_{\Delta_\gamma(\zeta)} \star \hat{g}^{s/s_1}(s_1, h - \eta + (\Delta_\gamma(\zeta) + \Delta_1)/2; \widehat{\beta})\right\}\right], \quad (68)$$

where

$$\langle\Delta\rangle = \int_s^1 \mathrm{d}y\,\Delta(y),$$

$$[\Delta](x) = x\Delta(x) - \int_0^x \mathrm{d}y\,\Delta(y),$$

$$\hat{g}(x, h; \beta) = e^{xf(x,h;\beta)}, \quad (69)$$

$$\Delta_\gamma(\zeta) = \Delta_f + \zeta^2\gamma^2,$$

$$\Delta_f = 2\Delta_r - \Delta(s) - \Delta_g,$$

and $f$ is the Parisi function that satisfies [49]

$$\frac{\partial f}{\partial x} = \frac{1}{2}\frac{\mathrm{d}\Delta(x)}{\mathrm{d}x}\left[\frac{\partial^2 f}{\partial h^2} + x\left(\frac{\partial f}{\partial h}\right)^2\right], \quad (70)$$

$$f(1, h; \widehat{\beta}) = \ln g_{\Delta(1)}(1, h; \widehat{\beta}).$$

If we take the limit $s \to 0$ we get back the Monasson replicated free energy of Eq. (92). Moreover taking the derivative with respect to $s$ and sending $s \to 0$ we get the Franz-Parisi potential that gives the free energy of a typical glassy state planted at $(\widehat{\varphi}_g, \widehat{\beta}_g)$ once followed to $(\widehat{\varphi}, \widehat{\beta})$. This is given by

$$-\widehat{\beta}\overline{f\left[\alpha(\widehat{\varphi}_g, \widehat{\beta}_g), \widehat{\beta}, \widehat{\varphi}, \gamma\right]}^{\alpha(\widehat{\varphi}_g, \widehat{\beta}_g)} = \text{const} + \frac{d}{2}\log\left(\frac{\pi\langle\Delta\rangle}{d^2}\right) - \frac{d}{2}\int_0^1 \frac{\mathrm{d}y}{y^2}\log\left(\frac{\langle\Delta\rangle + [\Delta](y)}{\langle\Delta\rangle}\right)$$
$$+ \frac{d}{2}\frac{m\Delta_f + \Delta_g}{m\langle\Delta\rangle} + \frac{d\widehat{\varphi}_g}{2}\int_{-\infty}^{\infty}\left[\mathscr{D}\zeta\int_{-\infty}^{\infty}\mathrm{d}h\,e^h g_{\Delta_g}^m\left(1, h + \Delta^g/2; \widehat{\beta}_g\right)\right.$$
$$\left. \times \int_{-\infty}^{\infty}\mathrm{d}x'\,f(0, x' + h - \eta + \Delta(0)/2; \widehat{\beta})\frac{e^{-\frac{1}{2\Delta_\gamma(\zeta)}(x' - \Delta_\gamma(\zeta)/2)^2}}{\sqrt{2\pi\Delta_\gamma(\zeta)}}\right]. \quad (71)$$

where $\mathscr{D}\zeta = \mathrm{d}\zeta e^{-\zeta^2/2}/\sqrt{2\pi}$. At this point we have to write the saddle point equations for the order parameter $\Delta_{ab}$. First of all it is quite simple to see that the saddle point equation for $\Delta_g$ does not depend on $\Delta_r$ and $\Delta(x)$ in the limit $s \to 0$ since the master replicas are completely uncorrelated from the slave ones. Indeed $\Delta_g$ is fixed by Eq. (58). The saddle point equations for $\Delta_r$ and $\Delta(x)$ can be obtained following the same strategy of [39] and they are shown in Appendix D. These equations coincide with the ones obtained in [39] the only difference being $f(1, h; \widehat{\beta})$. Starting from them it is can be shown that when $\dot{\Delta}(x) \neq 0$ the relation

$$0 = -1 + \frac{\widehat{\varphi}_g}{2}\int_{-\infty}^{\infty}\mathrm{d}h\,P(x, h)\left[G(x)f''(x, h; \widehat{\beta})\right]^2 \quad (72)$$

holds for all values of $x$ for which $\dot{\Delta}(x) \neq 0$. This equation is crucial to detect the Gardner transition point. Indeed, when evaluated on a 1RSB ansatz at $x = 1$, it gives the limit of stability of the replica symmetric solution.

## 6.2 The saddle point equations for equilibrium glasses starting from the normal glass phase

In this section we consider the case in which the glass that is prepared at $(\widehat{\varphi}_g, \widehat{\beta}_g)$ is at equilibrium meaning that $m = 1$. In this case the 1RSB solution for the planted glass is stable, see Fig. 3, and thus we can expect that if we apply a very small perturbation, the state will remain stable. Thus we can look for a 1RSB solution of the saddle point fullRSB equations. In this case there are only two parameters to be fixed that are $\Delta_r$ and $\Delta(x) = \Delta$ for $x \in [0, 1]$ where $\Delta$ is a constant. The free energy of the glass planted at $(\widehat{\varphi}_g, \widehat{\beta}_g)$ and then followed at $(\widehat{\varphi}, \widehat{\beta}, \gamma)$ is given by

$$
-\widehat{\beta} \overline{f\left[\alpha(\widehat{\varphi}_g, \widehat{\beta}_g), \widehat{\beta}, \widehat{\varphi}, \gamma\right]}^{\alpha(\widehat{\beta}_g, \widehat{\varphi}_g)} = \text{const} + \frac{d}{2} \frac{2\Delta_r - \Delta}{\Delta} + \frac{d}{2} \log(\Delta)
$$
$$
+ \frac{d\widehat{\varphi}_g}{2} \int_{-\infty}^{\infty} \mathscr{D}\zeta \int_{-\infty}^{\infty} dh \, e^h g_{2\Delta_R(\zeta) - \Delta}\left(1, h + \Delta_R(\zeta) - \Delta/2; \widehat{\beta}_m\right) \log g_\Delta\left(1, h - \eta + \Delta/2; \widehat{\beta}_s\right),
$$
(73)

where $\Delta_R(\zeta) = \Delta_r + \zeta^2 \gamma^2/2$. Taking the variational equations with respect to $\Delta$ and $\Delta_r$ we can reduce the fullRSB equations to their 1RSB counterpart. They are given by

$$
\frac{2\Delta_r}{\Delta^2} - \frac{1}{\Delta} = \widehat{\varphi}_g \int_{-\infty}^{\infty} \mathscr{D}\zeta \int_{-\infty}^{\infty} dh \, e^h \frac{\partial}{\partial \Delta} \left[g_{2\Delta_R(\zeta) - \Delta}\left(1, h + \Delta_R(\zeta) - \Delta/2; \widehat{\beta}_g\right)\right.
$$
$$
\left. \times \log g_\Delta\left(1, h - \eta + \Delta/2; \widehat{\beta}\right)\right]
$$
$$
0 = \frac{2}{\Delta} + \widehat{\varphi}_g \int_{-\infty}^{\infty} \mathscr{D}\zeta \int_{-\infty}^{\infty} dh \, e^h \left[\frac{\partial}{\partial \Delta_r} g_{2\Delta_R(\zeta) - \Delta}\left(1, h + \Delta_R(\zeta) - \Delta/2; \widehat{\beta}_g\right)\right]
$$
$$
\times \log g_\Delta\left(1, h - \eta + \Delta/2; \widehat{\beta}\right)
$$
(74)

For $\gamma = 0$ we get back the equations that were studied in [27]. In the limit $\widehat{\beta}_g \to \infty$ and $\widehat{\beta} \to \infty$ we get back the state following equations that were studied in [38]. These equations can be solved numerically for a specific choice of the interaction potential $\hat{v}(h)$ and we will do that in the case of Harmonic Soft Spheres.

## 6.3 The stability of the normal glass phase under external perturbation: the Gardner transition

The correctness of the 1RSB solution of the state following equations can be checked by looking at its local stability.[4] This can be done following the same line of reasoning of [46], Sec. 12. This gives access to the replicon eigenvalue that tests directly the stability of the solution. Here however we follow a simpler strategy, described in the context of the Sherrington-Kirkpatrick model in [50, 51] and in the random perceptron in [52], to obtain the Gardner point starting directly from the fullRSB solution. Let us suppose that there is a Gardner transition so that the 1RSB solution is unstable. Beyond that point, the correct solution is a solution where $\Delta(x)$ is no more costant and thus it is described by the equations of the previous section. A striking consequence of those equations is Eq. (72). Eq. (72) holds for all $x$ such that $\dot{\Delta}(x) \neq 0$. Coming from the fullRSB phase and approaching the Gardner transition, $\Delta(x)$ should go to a constant $\Delta$. This means that Eq. (72) holds up to the Gardner point where $\Delta(x)$ converges to its 1RSB value. In the stable glass phase, the same relation is not satisfied anymore due to the

---

[4]We underline here that we are focusing only on replica symmetry breaking instabilities. However there are also other kinds of instabilities as for example spinodal points, that do not involve RSB. This is the case of the yielding transition.

fact that $\dot{\Delta}(x) = 0$. This means that the Gardner transition appears when Eq. (72), evaluated on a 1RSB solution, is satisfied. Thus evaluating Eq. (72) on the solution of Eq. (74) we get the condition for the Gardner point

$$0 = -1 + \frac{\widehat{\varphi}_g}{2}\Delta^2 \int_{-\infty}^{\infty} \mathscr{D}\zeta \int_{-\infty}^{\infty} \mathrm{d}h\, e^{h+\eta-\frac{\Delta}{2}} g_{\Delta_\rho}\left(1, h+\eta-\frac{\Delta-\Delta_\rho}{2}, \widehat{\beta}_g\right)\left[\frac{\partial^2}{\partial h^2}\ln g_\Delta(1, h, \widehat{\beta})\right]^2$$

(75)

being $\Delta_\rho = 2\Delta_R(\zeta) - \Delta$. In the following we will compute the stability of the 1RSB solution within different state following protocols. Eq. (75) cannot tell what is the nature of the phase beyond the instability and as stated above we will assume that a marginal glass phase appears.

# 7 The phase diagram of Harmonic Soft Spheres

The formalism that we have developed in the previous sections allows us to obtain the phase diagram of thermal glasses formed by harmonic spheres in the limit of infinite dimension. The only thing we have to set is the specific function $g_\Delta$ in which it enters the interaction potential $\hat{v}$:

$$\begin{aligned}
g_\Lambda(1, h; \widehat{\beta}) &= \int \frac{\mathrm{d}y}{\sqrt{2\pi\Lambda}} \exp\left[-\frac{y^2}{2\Lambda} - \widehat{\beta}\hat{v}(y-h)\right] \\
&\equiv \int \frac{\mathrm{d}y}{\sqrt{2\pi\Lambda}} \exp\left[-\frac{y^2}{2\Lambda} - \frac{\widehat{\beta}(h-y)^2}{2}\theta(y-h)\right].
\end{aligned}$$

(76)

By solving the equations discussed in Sec. 4 we obtain the dynamical transition line and the phase diagram reported in Fig. 2. In the limit of infinite dimension the Mode-Coupling transition changes nature and instead of being a cross-over it becomes a true transition. At high temperature and low density the system relaxes on times of order one. Approaching the dynamical line the time-scale diverges and below it the system can be trapped in one out of many stable amorphous solids whose life-time is infinite in the $d \to \infty$ limit.

This phase diagram is similar to the one obtained by approximate means in [53] for three dimensional elastic spheres and coincides at zero temperature with the one already obtained for hard spheres [17].

In the following we will consider an amorphous solid formed at packing fraction and inverse temperature $(\widehat{\varphi}_g, \widehat{\beta}_g)$. In a realistic situation, this would mean that the cooling rate is such that the glass transition takes place at $(\widehat{\varphi}_g, \widehat{\beta}_g)$.[5] We can show that in the infinite dimensional limit the initial equilibrium glass state is always a normal glass so that the state following equations admit at least at the beginning a 1RSB solution. This means that starting from the equilibrium glass we have to solve the Eqs. (74) and control the stability of the replicon eigenvalue given by Eq. (75) in order to check where the Gardner phase emerges.

As anticipated in the introduction, in the next sections we shall study the effect of three perturbations (temperature, density and shear strain) on the amorphous solids prepared at $(\widehat{\varphi}_g, \widehat{\beta}_g)$.

## 7.1 Perturbation I: Cooling

Here we consider the case in which we form (plant) a glass in the state point $(\widehat{\varphi}_g, \widehat{\beta}_g)$, we keep $\widehat{\varphi}_g$ fixed and we consider different values of $\widehat{\beta}_g$. We note that we can do that only if we are below the dynamical transition line of Fig. 2 where the supercooled liquid is indeed a superposition of glassy states. Starting from these well prepared glasses, we cool the system

---

[5]The extension to state following of non equilibrium states is straightforward.

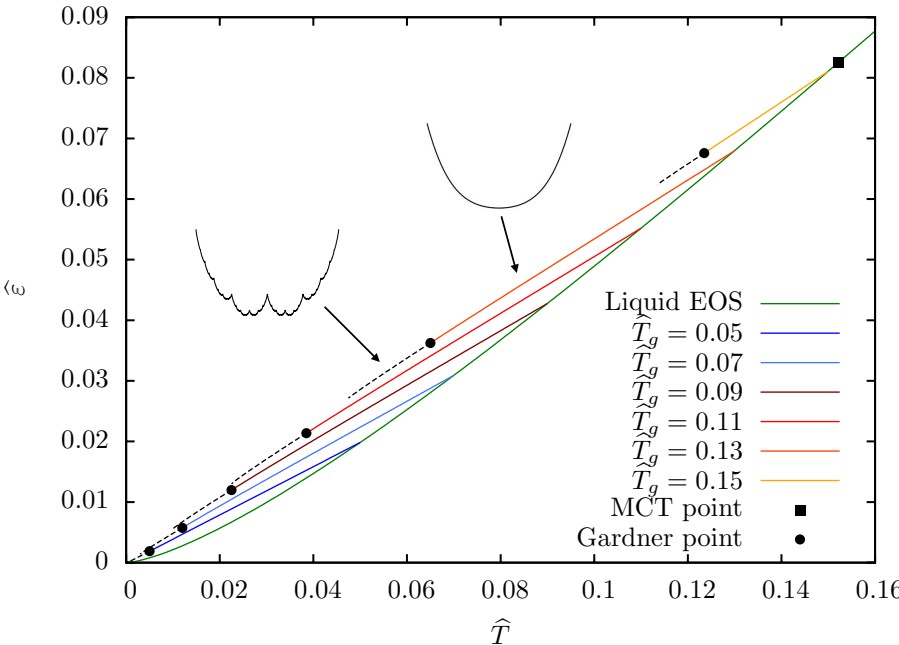

Figure 5: We plot the internal energy $\widehat{\varepsilon}$ (properly scaled with the dimension $d$) as a function of the temperature for different glasses planted at $(\widehat{\varphi}_g = 8, \widehat{T}_g)$ and then followed when the temperature is decreased. Picture reprinted from [27].

and check whether a Gardner transition takes place. In Fig. 5 we show the result for such annealing procedure with the corresponding Gardner transition points [27]. It is interesting to note that glassy states prepared at higher temperature tend to be much closer to the Gardner point with respect to well equilibrated low temperature glasses. The dashed lines correspond to the continuation of the 1RSB solution, which is only an approximation in the marginal phase.

In order to study the density dependence of the results found by cooling we use the following protocol. We prepare different equilibrium glasses at the same inverse temperature $\widehat{\beta}_g$ and with different initial packing fraction $\widehat{\varphi}_g$. Then we cool each one of these glasses. In Fig. 6 we show the phase diagram in this case. Again, we find that well equilibrated glasses, very far from the dynamical (mode-coupling) transition point tend to be much more stable than glasses close to the MCT point.

This is the major conclusion of the analysis on cooling: not very stable, i.e. not very deep, glasses are more prone to undergo a Gardner transition when lowering the temperature. This opens the possibility that very stable glasses formed at low enough temperature do not display a Gardner transition [54].

## 7.2 Perturbation II: Compression

In this case we prepare a glass in the state point $(\widehat{\varphi}_g, \widehat{\beta}_g)$ and we start to compress the system, namely increasing $\eta$ and keeping the inverse temperature $\widehat{\beta}_g$ fixed so that $\widehat{\beta} = \widehat{\beta}_g$. The general phase diagram in this case is in Fig. 7. We note that there are two Gardner transition lines. In order to discuss them let us start from what happens at zero compression, namely for $\eta = 0$. In this case, since $\widehat{\varphi} = \widehat{\varphi}_g$, there exist a temperature, the dynamical point, where glassy states disappear. At that point the replicon eigenvalue, which controls the instability toward the Gardner phase, is strictly equal to zero. If we increase $\eta$, for temperatures slightly smaller

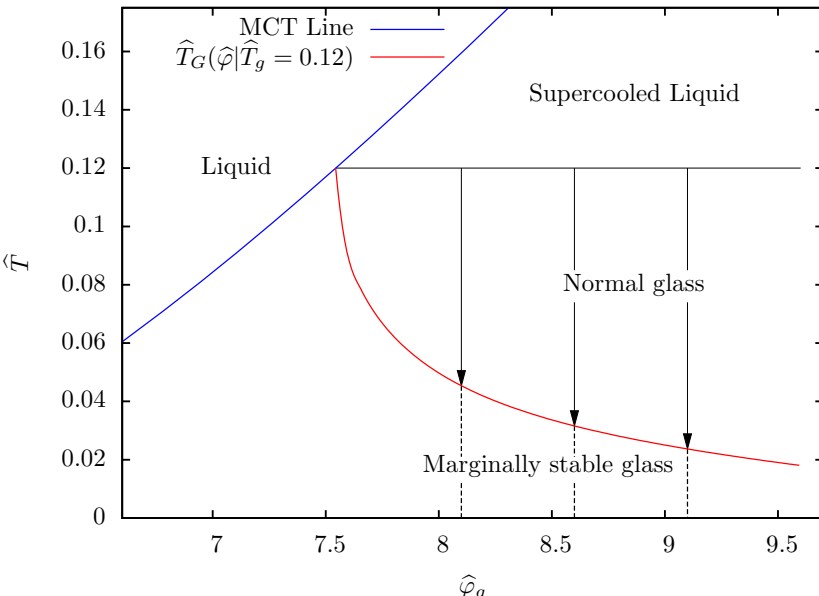

Figure 6: The Gardner transition line for glasses that are planted at $(\widehat{\varphi}_g, \widehat{T}_g = 0.12)$. Again we see that well equilibrated glasses undergo the Gardner transition at lower temperatures. Picture reprinted from [27].

than the dynamical one, we expect to find just by continuity a zero replicon for very small compressions. This is indeed what happens. Upon compression, glasses very close to the dynamical temperature tend to be very unstable and to undergo a Gardner transition very soon. Upon decreasing the planting temperature $\widehat{T}_g$, we see that this Gardner transition line can be followed up to zero temperature and it ends on a precise point at $\eta^*$. We call this Gardner transition line, the thermal Gardner transition. Beyond this line, for higher packing fractions each glass is supposed to be marginally stable and in order to obtain what happens for sufficiently high values of $\eta$ we need to compute the fullRSB solution (we leave this for future work).

As shown in Fig. 7, there is a small island where the Gardner phase emerges which is around the jamming point of hard spheres. Indeed, let us now consider a glassy state of soft spheres prepared exactly a zero temperature $\widehat{T}_g = 0$. Our protocol would then correspond to compress a glass of hard spheres. In this case, we know from [38] that hard spheres undergo a Gardner transition before reaching the jamming point. This is the point $\eta^-$ in Fig. 8 which coincides with what has been found in [38]. Compressing further, the system enters in a marginal glass phase and then jams at $\eta_J$. For soft spheres, one can keep compressing generating zero temperature states with non zero energy. In this case we find that the system goes back from a marginal glass phase to a normal glass phase at $\eta^+$ (the $\widehat{T}_g \to 0$ limit can be discussed analytically, see appendix B). By continuity, if we switch on a very small temperature, the points $\eta^-$ and $\eta^+$ get shifted. As shown in Fig. 8, this leads to a Gardner transition line.

The major outcome from the analysis on compression is that the Gardner transition takes place whenever the system is pushed toward an instability. Indeed it takes place before the spinodal and in vicinity of the jamming transition where the system is marginally stable (leading to a an island of marginal glasses surrounding the jamming point). This is crucial to ensure the correct criticality at the jamming transition [46] and clearly shows that the jamming transition point is very special.

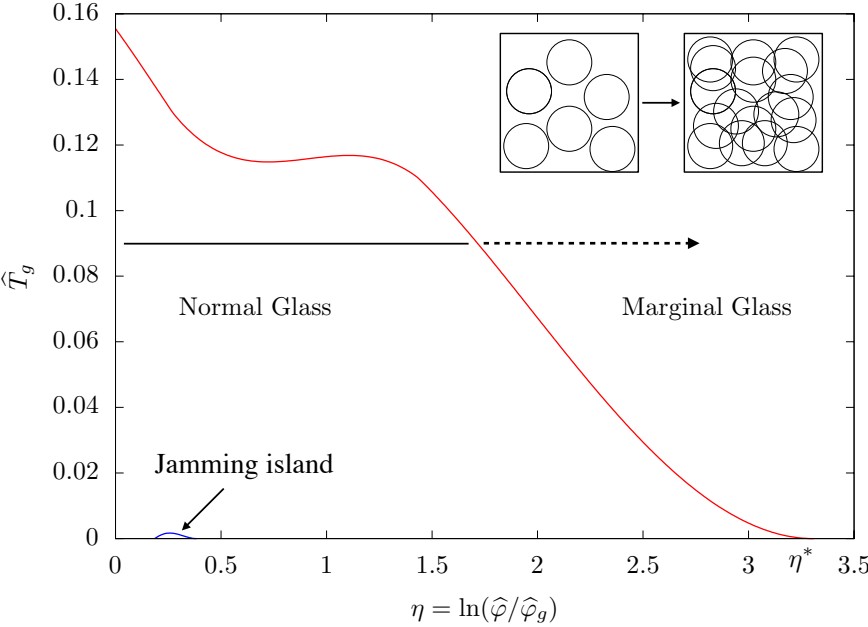

Figure 7: The phase diagram of glasses prepared at $(\widehat{\varphi}_g = 8, \widehat{T}_g)$ and then compressed. There are two Gardner transition lines. The red one is the thermal Gardner transition. The blue one instead surrounds the jamming point and is crucial to ensure that the behavior of soft sphere packing close to jamming is critical and characterized by the critical exponents computed in [16].

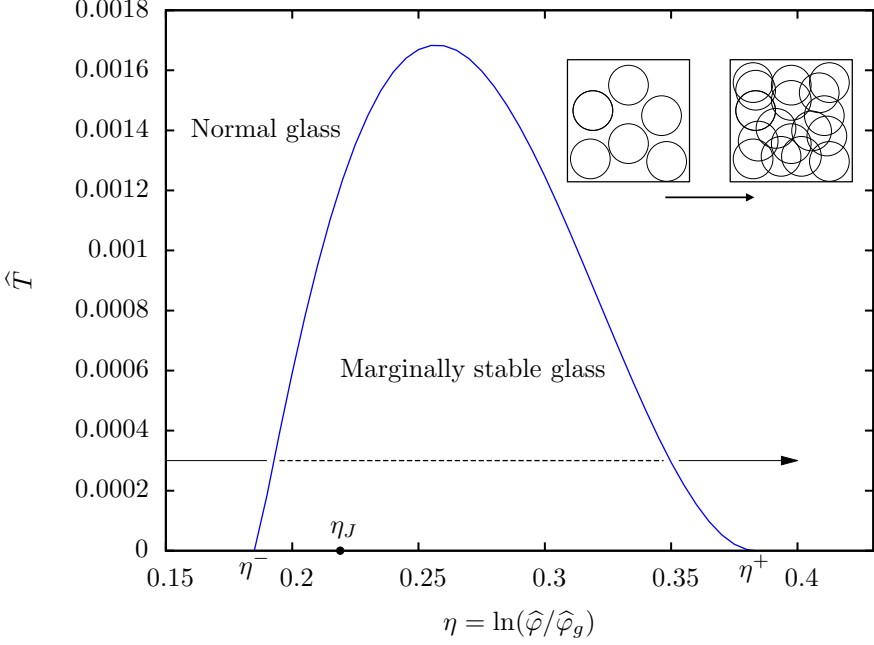

Figure 8: The Gardner transition line that surrounds the jamming point at $\widehat{\varphi}_g = 8$. The island of marginally stable glasses is what appears in Fig 7. Picture reprinted from [27].

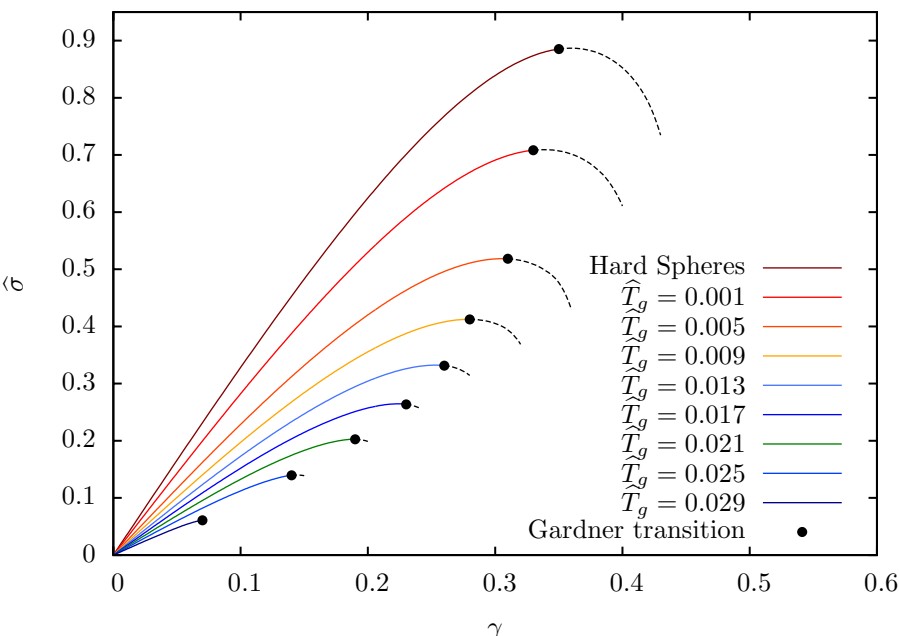

Figure 9: The equation of state of different glasses prepared at $(\widehat{\varphi}_g = 6, \widehat{T}_g)$ and followed when a shear strain is applied. For sufficiently strong strain each glass undergoes a Gardner transition (black dots) and then a yielding transition. The parts of the curves right after the Gardner transitions are just an approximation since they are computed within a 1RSB ansatz which is unstable.

## 7.3 Protocol III: Shearing

Here we study the case in which one imposes to a glass prepared at $(\widehat{\varphi}_g, \widehat{\beta}_g)$ a strain $\gamma$ as in (20). The corresponding phase diagram was anticipated in Fig.1. One finds that increasing the strain glasses first undergo a Gardner transition and then a yielding transition.

In Fig. 9 we plot the stress as a function of the strain for glasses prepared at the same packing fraction $\widehat{\varphi}_g = 6$ and different temperatures $\widehat{T}_g$. The stress $\widehat{\sigma}$ is obtained within the 1RSB ansatz by

$$
\widehat{\sigma} \equiv \frac{\widehat{\beta}\sigma}{d} = \frac{\widehat{\beta}}{d} \overline{\frac{\partial f\left[\alpha(\varphi_g,\beta_g),\beta,\varphi,\gamma\right]}{\partial \gamma}}^{\alpha(\varphi_g,\beta_g)} =
$$
$$
-\frac{\widehat{\varphi}_g \gamma}{2} \int_{-\infty}^{\infty} \mathcal{D}\zeta \, \zeta^2 \int_{-\infty}^{\infty} \mathrm{d}h \, e^h \frac{\partial}{\partial \Delta_r} g_{2\Delta_R(\zeta)-\Delta}\left(h + \Delta_R(\zeta) - \frac{\Delta}{2}; \widehat{\beta}_g\right) \ln g_\Delta\left(h - \eta + \frac{\Delta}{2}; \widehat{\beta}_g\right),
$$
$$(77)$$

where $\Delta_R(\zeta) = \Delta_r + \gamma^2 \zeta^2/2$. We also plot the Hard Sphere case that corresponds to $\widehat{T}_g = 0$ and that has been studied in [38,39]. As it can be seen from the curves, the shear modulus at $\gamma = 0$ is a decreasing function of the temperature: colder glasses are more rigid than hotter ones. Moreover, when the temperature is decreased the curves display an overshoot that becomes more and more pronounced as the temperature is lowered. We observe that for sufficiently high strain, each glass undergoes a Gardner transition. Beyond that point each glass enters in a marginally stable phase. The 1RSB solution is just an approximation in that regime (dashed line), to obtain a fully correct description one would need to solve numerically the fullRSB equations [39] (we leave it for future work). As shown by the 1RSB solution, strained glasses become unstable at a spinodal point. At that point the 1RSB equations loose the glassy solution

where $\Delta$ and $\Delta_r$ are finite and the system enters in a liquid/plastic phase. Within the fullRSB solution we expect the same behavior: the stress-strain curves can be followed up to a spinodal point that corresponds to the mean field yielding transition [38, 39, 55].

The analysis performed in this section leads to a conclusion very analogous to the one of the previous section: the Gardner transition emerges in regimes where the system is approaching an instability. When shearing, this instability corresponds to the yielding transition for which it would be interesting to develop a fullRSB analysis and determine the universality class at least within mean-field theory (naively it could be the one of the Random Field Ising model as for other glassy critical points [56–60] but additional physical effects due to fullRSB could play a role).

# 8 Conclusions

In this work we have extended and generalized the methods and the results obtained for Hard Spheres to the case of general soft spheres thermal glasses.

We have found that, generically, structural glasses undergo a Gardner transition towards a marginal glass phase when strongly perturbed. We have shown that this is the case when glasses are compressed, cooled and strained. By focusing on Harmonic Soft Spheres glasses we have been able to connect our results to Hard-Spheres ones. In particular we have found that the jamming point is very special in the temperature-density phase diagram, since it is surrounded by a marginally stable (Gardner) phase. The two main lessons we draw from the infinite dimensional results are that: (1) the Gardner transition is more likely to emerge when the system is pushed toward an instability (e.g. by shearing or compression), (2) the Gardner transition is favored when the initial glass state is less stable, e.g. closer to the jamming point where it is marginally stable and closer to the dynamical transition. Establishing whether three dimensional thermal glasses display a Gardner transition (or at least a remnant of it) is a crucial open issue [54]. Hopefully, our results will be useful guidelines for this line of research. Another important outcome of the present work is a greatly simplified derivation of the replicated free energy in the case of a generic interaction potential. This allows to treat shear strain deformations in thermal glasses and opens the way to study soft glassy rheology from first principles in the mean field limit. Moreover, the marginality of the Gardner phase brings about strong non linear responses and avalanches and thus the formalism we developed is the starting point to investigate them from first principles [27, 61].

# Acknowledgments

We thank L. Berthier, C. Scalliet and F. Zamponi for useful discussions. We acknowledge support from the ERC grants NPRGGLASS and from the Simons Foundation (N. 454935, Giulio Biroli). This work is supported by "Investissements d'Avenir" LabEx PALM (ANR-10-LABX-0039-PALM) (P. Urbani).

# A  Evaluation of $d$ dimensional angular integrals

We want to evaluate

$$\mathscr{A}_d \equiv \frac{d^{\frac{a+b}{2}}}{\Omega_d} \int d\theta_d f_1^a(\theta_d) f_2^b(\theta_d). \tag{78}$$

We can use as usual Gaussian integrals. We get

$$\mathscr{A}_d = \frac{d^{\frac{a+b}{2}}}{\mathscr{N}_d} \int \frac{\mathrm{d}^d x}{(\sqrt{2\pi})^d} x_1^a x_2^b e^{-|x|^2/2}, \qquad \mathscr{N}_d = \int \frac{\mathrm{d}^d x}{(\sqrt{2\pi})^d} |x|^{a+b} e^{-|x|^2/2}, \tag{79}$$

from which it follows that $\mathscr{A}_d = 0$ unless $a$ and $b$ are even. In the following we will assume that this is the case. Moreover we have that

$$\frac{\Omega_d}{(\sqrt{2\pi})^d} = \left[ \int_0^\infty \mathrm{d}x\, x^{d-1} e^{-x^2/2} \right]^{-1} \tag{80}$$

and moreover that

$$\int_0^\infty \mathrm{d}x\, x^{a-1} e^{-x^2/2} = 2^{(a-2)/2} \Gamma\left(\frac{a}{2}\right). \tag{81}$$

Using this we get

$$\begin{aligned}
\mathscr{A}_d &= d^{\frac{a+b}{2}} \frac{(\sqrt{2\pi})^d}{\Omega_d} (a-1)!!(b-1)!! \left[ \int_0^\infty \mathrm{d}x\, x^{d+a+b-1} e^{-x^2/2} \right]^{-1} \\
&= d^{\frac{a+b}{2}} (a-1)!!(b-1)!! \frac{\Gamma(d/2)}{2^{(a+b)/2}\Gamma((d+a+b)/2)}.
\end{aligned} \tag{82}$$

Since

$$\lim_{d\to\infty} d^{\frac{a+b}{2}} \frac{\Gamma(d/2)}{2^{(a+b)/2}\Gamma((d+a+b)/2)} = 1, \tag{83}$$

we get that

$$\lim_{d\to\infty} \mathscr{A}_d = \begin{cases} 0 & \text{if } a \text{ and } b \text{ are not even} \\ (a-1)!!(b-1)!! & \text{otherwise}. \end{cases} \tag{84}$$

## B  Compression at zero temperature

We can discuss in a simple way the case in which we compress the system starting from $\widehat{T}_g = \widehat{T} \to 0$. In this limit we have that

$$\Delta = v\widehat{T}_g, \tag{85}$$

while we expect that $\Delta_r$ stays of order one. The replicated free energy is thus given by

$$s = \text{const} + \frac{2\widehat{\beta}}{d} \left[ \frac{2\Delta_r}{v} - \frac{1}{1+v}\tilde{C}(\Delta_r) \right], \tag{86}$$

where we have defined

$$\tilde{C}(\Delta_r) = \frac{\widehat{\varphi}_g}{2} \int_{-\infty}^\infty \mathrm{d}h\, e^h \Theta\left[ \frac{h+\Delta_r}{\sqrt{4\Delta_r}} \right] (h-\eta)^2 \theta(\eta-h). \tag{87}$$

We can easily write the saddle point equations for $v$ and $\Delta_r$ that are given by

$$\begin{aligned}
\frac{2\Delta_r}{v^2} &= \frac{1}{(1+v)^2}\tilde{C}(\Delta_r), \\
\frac{2}{v} &= \frac{1}{1+v}\frac{\partial}{\partial\Delta_r}\tilde{C}(\Delta_r).
\end{aligned} \tag{88}$$

At this point we have only to compute the replicon eigenvalue. Taking Eq. (75), setting $\gamma = 0$ and taking properly the zero temperature limit we get that the Gardner transition happens when the following condition is satisfied

$$0 = -1 + \frac{\widehat{\varphi}_g}{2} \left( \frac{v}{1+v} \right)^2 \int_{-\infty}^{\eta} dh\, e^h \Theta \left[ \frac{h + \Delta_r}{\sqrt{4\Delta_r}} \right]. \tag{89}$$

The previous equations can be simplified. Indeed the saddle point equations can be rewritten as

$$\frac{2}{\Delta_r} = \frac{\left( \tilde{C}'(\Delta_r) \right)^2}{\tilde{C}(\Delta_r)}\,, \tag{90}$$
$$\frac{2\Delta_r}{\tilde{C}(\Delta_r)} = \left( \frac{v}{1+v} \right)^2,$$

where we have defined $\tilde{C}'(\Delta_r) = \partial \tilde{C}(\Delta_r)/\partial \Delta_r$. The equation on $\Delta_r$ is closed and can be solved numerically. Moreover we have that the replicon eigenvalue can be written in terms of the saddle point solution for $\Delta_r$ and it is given by

$$0 = -1 + \frac{\widehat{\varphi}_g \Delta_r}{\tilde{C}(\Delta_r)} \int_{-\infty}^{\eta} dh\, e^h \Theta \left[ \frac{h + \Delta_r}{\sqrt{4\Delta_r}} \right]. \tag{91}$$

The solution of these equations gives us the points $\eta_+$ and $\eta^*$ of Figs. 7 and 8.

## C  FullRSB equations for the planted glass phase

The fullRSB saddle point equations that describe the planted glass state at $(\widehat{\varphi}_g, \widehat{\beta}_g)$ can be derived following exactly the same strategy of [46] and here we will not give more details on that. The final equations are the same as Eq. (116) of [46], the only difference being in the initial condition for $f(1,h)$ that here is replaced by

$$f(1,h; \widehat{\beta}_g) = \ln g_{\Delta(1)}(1, h; \widehat{\beta}_g). \tag{92}$$

Furthermore, in the zero temperature limit of the Harmonic soft sphere case, they give back the same saddle point equations of the Appendix of [46]. Starting from these equations it can be shown that whenever $\Delta(x)$ has a continuous (not flat) part where $\dot{\Delta}(x) \neq 0$ the following equation is satisfied

$$0 = -1 + \frac{\widehat{\varphi}_g}{2} e^{-\Delta(m)/2} \int_{-\infty}^{\infty} dh\, P(x,h) \left( G(x) f''(x, h; \widehat{\beta}_g) \right)^2. \tag{93}$$

This will be important to establish the stability of the 1RSB solution. Indeed the relation evaluated at $x = 1$ and on a 1RSB profile for $\Delta(x)$ gives the stability of the normal glass phase. Finally, these equations, reduced to the 1RSB ansatz give Eq. (58).

## D  FullRSB saddle point equations for the state following protocol

In this section we summarize the fullRSB equation in the state following approach. They can be derived using the same strategy as [39] being the only difference the initial condition for

$f$.

$$G(x) = x\Delta(x) + \int_x^1 dz\,\Delta(z) \quad \Longleftrightarrow \quad \Delta(x) = \frac{G(x)}{x} - \int_x^1 \frac{dz}{z^2} G(z)\,,$$

$$f(1,h;\widehat{\beta}) = \log\gamma_{\Delta(1)} \star e^{-\widehat{\beta}\hat{v}(h)} = \log g_{\Delta(1)}(1,h;\widehat{\beta})$$

$$\frac{\partial f}{\partial x} = \frac{1}{2}\frac{\dot{G}(x)}{x}\left[\frac{\partial^2 f}{\partial h^2} + x\left(\frac{\partial f}{\partial h}\right)^2\right]\,,$$

$$P(0,h) = e^{h+\eta-\Delta(0)/2}\int_{-\infty}^{\infty}\mathscr{D}\zeta\int_{-\infty}^{\infty}\frac{dx}{\sqrt{2\pi\Delta_\gamma(\zeta)}}e^{-\frac{1}{2\Delta_\gamma(\zeta)}\left(x+\frac{\Delta_\gamma(\zeta)}{2}\right)^2}\,,$$

$$\times g_{\Delta_g}^m\left(1,h+\eta-x+(\Delta_g-\Delta(0))/2;\widehat{\beta}_g\right)\,,$$

$$\dot{P}(x,h) = -\frac{\dot{G}(x)}{2x}\left[P''(x,h) - 2x(P(x,h)f'(x,h;\widehat{\beta}))'\right]\,,$$

$$\frac{1}{G(0)} = -\frac{\widehat{\varphi}_g}{2}\int_{-\infty}^{\infty}dh\,P(0,h)\left(f''(0,h;\widehat{\beta}) + f'(0,h;\widehat{\beta})\right)\,,$$

$$\frac{m\Delta_f+\Delta_g}{mG(0)^2} = \frac{\widehat{\varphi}_g}{2}\int_{-\infty}^{\infty}dh\,P(0,h)\left(f'(0,h;\widehat{\beta})\right)^2\,,$$

$$\kappa(x) = \frac{\widehat{\varphi}_g}{2}\int_{-\infty}^{\infty}dh\,P(x,h)\left(f'(x,h;\widehat{\beta})\right)^2\,,$$

$$\frac{1}{G(x)} = \frac{1}{G(0)} + x\kappa(x) - \int_0^x dy\,\kappa(y) \qquad x > 0$$

(94)

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
