# Peer review of "Liu-Nagel phase diagrams in infinite dimension"

_SciPost Physics, doi:SciPost Phys. 4, 020 (2018)_

## Round 1 · Referee Report · Anonymous · 2017-8-21

Strengths
1. The results presented in this paper are very interesting and will be potentially very important to understand the long reported anomalous behavior of amorphous materials and to rationalize the experimental response of disordered solids to externa stresses.
2. This article presents a detailed and simple description of the exact derivation of the Liu-Nagel diagrams in infinite dimensions for harmonic soft spheres, enabling to reproduce the results presented by the same authors in a recent paper [Nature Physics 12, 1130–1133 (2016)], something that would be extremely difficult without this technical paper.
3. Still, the results discussed here go beyond those presented in the previous paper, and extend to soft potentials results obtained recently for hard spheres in infinite dimensions, specially for the part concerning shearing and yielding.
Weaknesses
1. The figures concerning the phase diagrams (mostly fig 1, the Nagel-Liu diagram) are difficult to interpret. The axes are not the traditional ones, and I find hard to abstract their meaning.
2. Although the first part of the paper is easy to follow, the part concerning the fullRSB results (from section V) is completely obscure and hard to follow. There is a constant reference to previous papers for derivations, equations become lists of equations where the variables that appear are not even defined...
Report
As I indicated above in the Strengths section, I believe the paper is very interesting and will strongly contribute to the condense matter community, and for this reason, I cannot do other but recommend its publication om SciPost. However, I think that revisions are compulsory to make this paper more available to the community. I list below the requested changes.
Requested changes
General comments,
1. Fig 1 is hard to understand. The position of the liquid, supercooled liquid and the glass should be written in the figure. A cross-section surface of the figure, where the dynamical line becomes just a point, might help to understand the different transitions. Like it is now, I don't see a Gardner transition under cooling or compression.
2. I understand that the fullRSB derivations might be too long to include them in this paper, and considering that they are very similar to those presented in previous papers of hard spheres, the decision was to omit them and refer the previous papers. The problem is that, as they appear now, it's impossible to understand anything, I think it is only accessible to people familiar with the infinite dimensional hard-sphere calculations, which corresponds to a very reduced part of the community. I suppose the list of formulas concerning magnitudes not defined in the text aim to help a future reader trying to repeat the results, but in such a case, they would rather gather them in an appendix, as they are now they only contribute to confusion.
Now, minor comments,
3. From the beginning, the authors refer to the scaled packing fraction and temperature, unless I am wrong, I don't see the definition of this scaling anywhere.
4. Below Eq. (1), volume is defined as V, I think authors wanted to define N, the number of particles instead, I don't see V anywhere.
5. $s$ is not defined
6. I find a bit confusing the introduction of m, authors refer to the 'biased partition function' whose meaning is not discussed. Is it $\beta\to m \beta$?
7. The 1 in Eq. (13) shouldn't be a m?
8. I don't understand why an $Y$ does not appear in Eq. (33).
9. I find the reasoning below Eq. (39) very unclear.
10. The term $e^{h}$ that appears after the change of variable I suppose that arrives after taking the $d\to\infty$ limit, something should be mentioned.
11. At the end of Eq. (41) the a->c for the product.
12. What is (1) and (2) in Eq. (44).
13. I find hard to understand the justification for equations (45) and (46).

---

## Round 2 · Author Response

We thank the referee for her/his positive report and for having appreciated our work. We submit an improved presentation in which we took into account all her/his comments and suggestions.

Below we reply to her/his comments.

General comments:

  1. There is no Gardner transition in cooling or compression. In fact we strain (infinite dimensional) glasses that are prepared at equilibrium above the Kauzmann transition, thus they are always stable in terms of replica symmetry. We have underlined this in the caption of the figure and in a footnote. We have moreover change the text in the introduction and the caption to better clarify the figure. We tried to draw a surface to join the yielding transition points, as the referee suggested, but we found it was not improving the figure so we preferred to keep the old version.
  2. We moved the part on FRSB to appendices in order to clarify the presentation as suggested by the referee.
  3. The reduced temperature and packing fraction are defined in Eq.(17) and Eq.(41). We have underlined this in the caption of Fig.1.
  4. We fixed the typo. We thank the referee.
  5. S is defined in Eq.(11) as the number of replicas. The product runs from 1 to s.
  6. The referee is right. Its definition is in Eq.(8).
  7. We thank the referee. We corrected the typo.
  8. Y does not appear because first $\rho(y)$ does not depend on it, and second one can change variable from X+Y->X. The integration over Y produces the $1/\rho$ factor.
  9. In the limit of infinite dimensions the two last terms of (36) can be evaluated using the central limit theorem since they both correspond to a sum of an infinite number of terms. This is the reasoning we use for eq. 37 and eq. 39. We have added a sentence below eq. 36 to stress this point.
  10. We did as suggested.
  11. We fixed the notation.
  12. We have modified the text skipping the previous equations and we have underlined that the equation (48) can be derived using the same kind of computations in of Eq. (38) and (39).

You are currently on this page

Resubmission 1704.04649v2 on 17 January 2018

---

## Editorial Decision

published